# Induced seismicity in geologic carbon storage

**Víctor Vilarrasa[1,2], Jesus Carrera[1,2], Sebastià Olivella[3], Jonny Rutqvist[4] and Lyesse Laloui[5]**

[1] Institute of Environmental Assessment and Water Research, Spanish National Research Council (IDAEA-CSIC), Barcelona, Spain

[2] Associated Unit: Hydrogeology Group (UPC-CSIC), Barcelona, Spain

[3] Dept. of Civil and Environmental Engineering, Technical University of Catalonia (UPC-BarcelonaTech), Barcelona, Spain

[4] Energy Geosciences Division, Lawrence Berkeley National Laboratory, CA, USA

[5] Laboratory of Soil Mechanics, École Polytechnique Fédérale de Lausanne (EPFL), Lausanne, Switzerland

*Correspondence to*: Victor Vilarrasa (victor.vilarrasa@idaea.csic.es)

**ABSTRACT**

Geologic carbon storage, as well as other geo-energy applications, such as geothermal energy, seasonal natural gas storage and subsurface energy storage, imply fluid injection/extraction that causes changes in rock stresses field and may induce (micro)seismicity. If felt, seismicity has a negative effect on public perception and may jeopardize wellbore stability and damage infrastructure. Thus, induced earthquakes should be minimized to successfully deploy geo-energies. However, numerous processes may trigger induced seismicity, which contributes to making it complex and translates into a limited forecast ability of current predictive models. We review the triggering mechanisms of induced seismicity. Specifically, we analyze (1) the impact of pore pressure evolution and the effect that properties of the injected fluid have on fracture/fault stability; (2) non-isothermal effects caused by the fact that the injected fluid usually reaches the injection formation at a lower temperature than that of the rock, inducing rock contraction, thermal stress reduction and stress redistribution around the cooled region; (3) local stress changes induced when low permeability faults cross the injection formation, which may reduce their stability and eventually cause fault reactivation; (4) stress transfer caused by seismic or aseismic slip; and (5) geochemical effects, which may be especially relevant in carbonate containing formations. We also review characterization techniques developed by the authors to reduce the uncertainty on rock properties and subsurface heterogeneity both for the screening of injection sites and for the operation of projects. Based on the review, we propose a methodology based on proper site characterization, monitoring and pressure management to minimize induced seismicity.

**Keywords:** $CO_2$ injection, pressure evolution, coupled processes, caprock integrity, fault reactivation

## 1. INTRODUCTION

The interest in subsurface energy resources, such as geologic carbon storage, geothermal energy and subsurface energy storage, has significantly increased as a means to mitigate climate change (IPCC, 2018). In particular, geologic carbon storage has the potential to store large amounts of carbon dioxide ($CO_2$) in deep geological formations, reducing $CO_2$ emissions to the atmosphere (Hitchon et al., 1999; Celia, 2017). Such subsurface energy-related activities imply fluid injection/extraction that change the pore pressure and thus, the effective stresses, causing deformation and potentially fracture and/or fault reactivation that may lead to induced (micro)seismicity (Ellsworth, 2013; Grigoli et al., 2017).

Induced microseismicity, i.e., seismicity of such low magnitude that is not felt on the ground surface (typically moment magnitude $M<2$), is positive if confined within the injection formation because shear slip of fractures enhances permeability (Yeo et al., 1998; Vilarrasa et al., 2011; Rutqvist, 2015). This permeability enhancement permits injecting the same amount of fluid at a lower injection pressure, thus reducing compression costs. However, induced microseismicity should be avoided in the caprock because its sealing capacity could be compromised, which could lead to $CO_2$ leakage. Additionally, if felt, induced earthquakes may damage wells, buildings and infrastructure and may cause fear and nuisance to the local population (Oldenburg, 2012). As a result of these negative effects, several geo-energy projects have been cancelled before they entered into operation, such as the enhanced geothermal systems (EGS) at Basel, Switzerland (Häring et al., 2008; Deichmann et al., 2014) and Pohang, South Korea (Grigoli et al., 2018; Kim et al., 2018), a hydrothermal project at Sankt Gallen, Switzerland (Edwards et al., 2015; Diehl et al., 2017) and a seasonal gas storage project at Castor, Spain (Cesca et al., 2014; Gaite et al., 2016). Thus, felt induced seismic events

have to be minimized, and ideally avoided, in order to achieve a successful deployment of geo-energy projects.

Geologic carbon storage projects, both at large scale and pilot scale, have not induced any felt earthquake to date (White and Foxall, 2016; Vilarrasa et al., 2019). This lack of felt seismicity may be due to some favorable aspects of $CO_2$ storage with respect to water injection that will be explained in this paper. Yet, induced microseismicity is common, such as at In Salah, Algeria (Stork et al., 2015; Verdon et al., 2015), Decatur, Illinois (Kaven et al., 2015; Bauer et al., 2016), and Otway, Australia (Myer and Daley, 2011), projects. Despite the absence of felt seismicity to date, proper protocols should be defined and followed to avoid inducing felt earthquakes in future geologic carbon storage projects.

The aim of this paper is to review the potential causes of induced seismicity in geologic carbon storage and to explain methodologies that can serve to minimize the risk of inducing felt seismic events. First, we introduce the potential triggering mechanisms of induced seismicity and then, we go into details of each of them. Specifically, we review the stress state of deep geological formations, the pore pressure evolution, non-isothermal effects resulting from $CO_2$ injection, shear slip stress transfer and geochemical effects on geomechanical properties and how these effects may lead to induced microseismicity. Afterwards, we analyze how $CO_2$ injection affects fault stability and, finally, we present subsurface characterization techniques that can be used to minimize the occurrence of felt induced seismicity.

## 2. TRIGGERING MECHANISMS

The basic principle of induced seismicity is that the pressure buildup caused by fluid injection reduces the effective stresses, which brings the stress state closer to failure (Figure 1). If failure conditions are reached, the elastic energy stored in the rock mass is released and a (micro)seismic event is induced. Failure in geomaterials can occur either in tensile or shear mode (Jaeger et al., 2009). While tensile failure induces microseismic events of such low magnitude that cannot be felt on the ground surface, shear failure may lead to felt earthquakes if a sufficiently large area of a pre-existing discontinuity, i.e., a fracture or fault, is reactivated. Nevertheless, in the cases in which tensile failure is sought, i.e., to create hydraulic fractures to enhance rock permeability, shear failure of pre-existing faults may also occur if they become pressurized during the hydraulic fracturing operations. In such situation, felt earthquakes associated to hydraulic fracturing operations may occur (Rubinstein and Mahani, 2015). For example, a felt earthquake occurred at the Preese Hall 1 exploration well for shale gas near Blackpool, UK, during hydraulic fracturing because a pre-existing nearby fault was reactivated (Clarke et al., 2014).

In principle, fluid pressure buildup may seem the only mechanism that induces seismicity. Thus, intuition suggests that stability should improve in the vicinity of the injection well after injection is stopped because fluid pressure drops rapidly. Far away from the injection well, fluid pressure continues to rise and thus, pressure-diffusion could explain continued post-injection induced seismicity (Hsieh and Bredehoeft, 1981), which is often observed after stimulation of EGS (Parotidis et al., 2004). However, pressure-diffusion cannot explain why the magnitude of post-injection seismicity is often higher than that induced during injection, e.g., at Basel, Switzerland (Deichmann and Giardini, 2009), at Soultz-sous-Forêts, France (Evans et al., 2005), and at Castor, Spain (Gaite et al., 2016). Even

though this high magnitude post-injection seismicity has not been observed in geologic carbon storage projects, its causes should be understood to prevent it. The counterintuitive occurrence of large magnitude post-injection induced seismicity may be explained by the fact that fluid injection in the subsurface involves coupled processes that are more complex than just the hydraulic effect:

- The stress state changes in response to pore pressure variations (Streit and Hillis, 2004; Rutqvist, 2012). Specifically, the total stress increases in the direction of flow due to the lateral confinement that opposes to the expansion of the rock in this direction (Zareidarmiyan et al., 2018). This poro-mechanical effect modifies the initial stress state and thus, the analysis of fault stability cannot be performed as a simple subtraction of the pressure buildup from the initial effective stress state;

- The injected $CO_2$ usually reaches the injection depth at a colder temperature than that of the rock because $CO_2$ does not reach thermal equilibrium with the geothermal gradient along its way down the well (Paterson et al., 2008). As a result, the storage formation cools down around the injection well, inducing a thermal stress reduction that brings the stress state closer to failure conditions (Vilarrasa and Rutqvist, 2017). The magnitude of induced thermal stresses is proportional to the rock stiffness. Thus, induced thermal stresses depend on the rock type in which fluid is injected, becoming larger in reservoir rocks than in clay-rich caprocks because reservoirs are usually stiffer (Vilarrasa and Makhnenko, 2017);

- The stress changes that arise in the storage formation and the caprock as a result of pressure buildup and cooling vary depending on the rock properties and the contrast between geological layers (Verdon et al., 2011);

- Each (micro)seismic event provokes a stress redistribution around the portion of the fracture or fault that undergoes shear slip (Okada, 1992). This stress transfer controls the distribution of aftershocks in natural seismicity (King et al., 1994) and may be the reason for observed rotations in the direction of the sheared faults in sequences of induced seismicity during stimulation of EGS (De Simone et al., 2017a);

- Not all the shear slip occurring in fractures or faults induces seismic events. Actually, shear slip may occur aseismically (Cornet et al., 1997). This aseismic slip may induce (micro)seismic events away from the slipped surface (Guglielmi et al., 2015);

- Geochemical reactions may alter the frictional strength of faults, which could lead to failure conditions if a fault is weakened;

- Heterogeneity in the rock type, strength of faults and the stress field, which may present local variations around faults (Faulkner et al., 2006), affect fault stability.

All these potential triggering mechanisms are usually neglected because pressure-diffusion is considered sufficient to explain induced seismicity. Though pore pressure diffusion alone may explain certain sequences of induced events (Shapiro et al., 2002), seismic sequences are usually more complex and imply a combination of several coupled processes. For example, cooling-induced stresses resulting from $CO_2$ entering the storage formation 45 ºC colder than the rock may explain part of the microseismicity detected at In Salah, Algeria (Vilarrasa et al., 2015). Another example is Weyburn, Canada, where the scarce microseismic events (around 200) that were induced in the caprock at the beginning of injection were interpreted to be caused by stress changes resulting from the contrast in stiffness between the reservoir and caprock (Verdon et al., 2011). Thus, when

assessing the potential for induced (micro)seismicity of $CO_2$ storage projects, all these coupled processes should be considered (Figure 2).

## 3. STRESS STATE

A careful examination of the subsurface stress state reveals that crystalline rocks accumulate more stress as a result of tectonics than sedimentary rocks (Vilarrasa and Carrera, 2015). The dependence of the stress state on the rock type reflects the contrast in the rock stiffness. Since crystalline rocks are much stiffer than sedimentary rocks, stresses induced by tectonics mainly accumulate in the crystalline basement. In contrast, the relatively soft sedimentary rocks deform without accumulating large stresses and as a result they do not usually become critically stressed. This is demonstrated in Table 1, which displays the estimated stress state at several $CO_2$ storage sites with the corresponding mobilized friction coefficient, $\mu_{mob}=\tan\phi'_{mob}$. where $\phi'_{mob}$ is the mobilized friction angle. $\phi'_{mob}$ is the angle that forms the tangent to the Mohr circle assuming no cohesion. Thus, if the mobilized friction coefficient is lower than the actual friction coefficient, which is generally equal to 0.6 (Barton, 1976), the rock is not critically stressed. Interestingly, the mobilized friction coefficient is lower than 0.6 for all the $CO_2$ storage sites included in Table 1. Since $CO_2$ will be stored in sedimentary basins, the less likely criticality of stress implies that a certain pressure buildup and cooling can be applied without reaching failure conditions (Figure 3). Yet, there may be cases of critically stressed sedimentary rocks, which may lead to unexpected seismicity if no stress measurements are performed. Therefore, mechanical characterization must be required at potential storage sites.

The stress state at each site should be measured in order to determine the maximum sustainable injection pressure and maximum cooling that would lead to a safe $CO_2$ storage (Rutqvist et al., 2007; Kim and Hosseini, 2014). Thus, stress measurements should be routinely performed during wellbore perforation, determining both the magnitude and orientation of the principal stresses (Cornet and Jianmin, 1995). The range of strikes and dips of potentially reactivated faults can be determined once the stress state is known (Morris et al., 1996). This exercise is crucial to identify faults that may induce large seismic events, to foresee an optimal design of the injection strategy and to define mitigation measures (e.g., Birkholzer et al., 2012; Buscheck et al., 2012; Dempsey et al., 2014) if induced seismicity is predicted to possibly occur above a predefined threshold.

## 4. PRESSURE EVOLUTION

The pressure evolution of $CO_2$ injection is favorable to achieve a long-term geomechanically stable situation. In contrast to water injection, which yields a linear increase of pressure with the logarithm of time when a continuous flow rate is injected (Theis, 1935), $CO_2$ leads to a peak at the beginning of injection followed by a relatively constant overpressure (Figure 4). Thus, pressure evolution is relatively easy to control in $CO_2$ injection operations, which should help to minimize induced (micro)seismicity (Vilarrasa and Carrera, 2015). Such pressure evolution has been observed in the field, at Ketzin, Germany (Henninges et al., 2011), numerically (e.g., Vilarrasa et al., 2010; Okwen et al., 2011) and analytically (Vilarrasa et al., 2013a).

The initial sharp increase in pore pressure is due not only to viscous forces opposing fluid displacement, but also to capillary forces caused by the desaturation around the injection well, which decreases the relative permeability to both $CO_2$ and water (Figure 4b).

However, once $CO_2$ fills the pores around the injection well (Figure 4c), the $CO_2$ relative permeability rises. Additionally, since $CO_2$ viscosity is one order of magnitude lower than that of brine, $CO_2$ can flow easily inside the storage formation, which leads to a constant or even a slight drop in overpressure (Figure 4a). This constant evolution of fluid pressure is maintained as long as the pressure perturbation does not reach a boundary. Once a boundary is reached, pressure will decrease in the presence of a constant pressure boundary and will increase in the presence of a low permeability boundary. The pressure evolution shown in Figure 4 is not affected by boundary effects because the pressure perturbation does not reach the outer boundary during the displayed injection time. This fluid pressure evolution induces the largest effective stress changes in the caprock at the beginning of injection, coinciding with the peak in pressure increase.

Maintaining the caprock integrity in the long-term is favored by two effects that tend to decrease overpressure inside the storage formation: (1) $CO_2$ dissolution into the resident brine, and (2) brine flow across the low-permeability formations that confine the storage formation, i.e., caprock and base rock (Vilarrasa and Carrera, 2015). On the one hand, when $CO_2$ dissolves into brine, fluid pressure decreases because the total fluid volume is reduced (Mathias et al., 2011a; Steele-MacInnis et al., 2012). As observed in natural analogues, the percentage of $CO_2$ that may eventually become trapped by dissolution can be as high as 90% in carbonate storage formations (Gilfillan et al., 2009). In the short-term, $CO_2$ dissolution can also be high in storage formations with high vertical permeability ($k>10^{-13}$ m$^2$) because of the formation of gravity fingers induced by the unstable situation of having a fluid of a higher density, i.e., $CO_2$-rich brine, above a fluid of lower density, i.e., the resident brine (Riaz et al., 2006; Hidalgo and Carrera, 2009; Pau et al., 2010). On the other hand, caprock permeability at the field scale is two to three orders of magnitude larger than that at the core scale as a result of the presence of fractures

and faults (Neuzil, 1994). Thus, resident brine of the storage formation can flow across the caprock and base rock, lowering the pressure buildup inside the storage formation. Though brine can flow through the caprock because single phase flow is not hindered by capillarity, $CO_2$ cannot because of the high $CO_2$ entry pressure of clay-rich formations (Benson and Cole, 2008).

To quantify the flow across the caprock in the long-term, let us assume a 100-m thick caprock with permeability of $10^{-18}$ m$^2$, water viscosity of $4 \cdot 10^{-4}$ Pa·s (assuming a temperature of 60 ºC) and a mean overpressure of 1 MPa distributed in a radial distance of 20 km. This scenario yields a flux across the caprock of $2.5 \cdot 10^{-11}$ m/s in an area of $1.26 \cdot 10^9$ m$^2$. Thus, the flow rate across the caprock is of 0.031 m$^3$/s, which is in the order of magnitude of industrial scale injection rates (in the order of 0.05 m$^3$/s for annual megaton injection), effectively lowering the pressure increase inside the storage formation.

## 5. NON-ISOTHERMAL EFFECTS

In addition to pressure increase, thermal effects are also relevant in geologic carbon storage because temperature changes induce thermal stresses that affect fracture stability (Vilarrasa and Rutqvist, 2017). $CO_2$ reaches the bottom of the injection well at a temperature lower than that of the storage formation because $CO_2$ flow within the well is isenthalpic (Pruess, 2006) and thus, it heats up at a lower rate than the geothermal gradient (Lu and Connell, 2008). As a result, the rock around injection wells cools down.

To illustrate the effect on fracture stability, we present the simulation results of cold $CO_2$ injection into a deep saline aquifer. Figure 5a displays the model setup with the initial and boundary conditions. The material properties are included in Table A1 in the Appendix.

The advance of the cooling front with respect to the $CO_2$ plume is retarded because the rock has to be cooled down (compare Figures 5b and 5c) (Bao et al., 2014; LaForce et al., 2015; De Simone et al., 2017b). Cooling mainly advances by advection in the reservoir, but it also extends into the lower portion of the caprock by conduction (Figure 5c). The extent of the cooling region can become of a few hundreds of meters after some decades of $CO_2$ injection at industrial scale rates, i.e., megaton injection (Vilarrasa et al., 2014). Thus, unless faults are present in the vicinity of the injection well, they will not be directly affected by cooling. Nevertheless, faults located far from the cooling region may undergo stability changes as a result of the contraction of the cooled rock, which causes changes in far-field stresses (Jeanne et al., 2014).

The cooling-induced rock contraction and thermal stress reduction shift the stress state towards shear failure conditions and, theoretically, tensile fractures could be formed if the tensile strength was reached (Luo and Bryant, 2010; Goodarzi et al., 2010; 2012; Gor et al., 2013). The temperature-induced stresses are not isotropic (Figure 6), and thus, the effect on fracture stability depends on the stress regime, i.e., normal faulting, strike-slip or reverse faulting (Vilarrasa, 2016). In general, fracture stability becomes more compromised in the reservoir than in the caprock, which may lead to injectivity enhancement while maintaining the caprock sealing capacity (Goodarzi et al., 2015; Vilarrasa et al., 2017a).

This favorable situation occurs especially in normal faulting stress regimes (Vilarrasa et al., 2013b; Kim and Hosseini, 2015). Figure 6 displays how stress variations induced in the reservoir and caprock as a result of cooling affect fracture stability in a normal faulting stress regime (i.e., vertical stress larger than horizontal stresses). Both the vertical and horizontal stresses decrease inside the reservoir within the cooled region. The stress reduction is proportional to the rock stiffness, the rock thermal expansion coefficient and

the temperature change. The vertical stress reduction within the reservoir causes a disequilibrium in this direction because the overburden on top of the reservoir remains constant, so that vertical stresses become smaller than the weight of the material above (Figure 6a). Thus, to satisfy stress equilibrium and displacement compatibility, an arch effect develops to support the weight of the material above, leading to a reduction of horizontal stresses within the reservoir and an increase in the lower portion of the caprock (Figure 6b). The net result of these stress changes is to: (1) bring the reservoir towards shear failure conditions (the Mohr circles shifts to the left and increases in size, Figure 6c), and (2) improve stability of the caprock by tightening it (the Mohr circle becomes smaller, Figure 6d). This contrast in stability between the reservoir and the caprock is highlighted in Figure 5d, which shows that plastic strain, i.e., strain that occurs because failure conditions have been reached, only takes place in the reservoir and not in the caprock (for details on the failure surface, see Vilarrasa and Laloui, 2015).

The situation is slightly different in a reverse faulting stress regime, where the vertical stress is the minimum principal stress (Vilarrasa, 2016). The cooling-induced increase of horizontal stress in the lower portion of the caprock causes the Mohr circle to increases in size (i.e., the deviatoric stress increases). Nevertheless, this increase is slight because of the high confinement in reverse faulting stress regimes. Still, shear failure may occur as a result of cooling. Similarly, the deviatoric stress is maintained in a strike slip stress regime (Vilarrasa, 2016), which may induce shear failure of pre-existing fractures, and thus, induced microseismicity, in the cooled region of the caprock, as was likely the case at In Salah, Algeria (Vilarrasa et al., 2015). These results highlight again the importance of characterizing the stress state.

The simulation results shown in Figures 5 and 6 consider that the thermal expansion coefficient of the storage formation and the caprock are equal. Despite the limited range

of the values that the thermal expansion coefficient can take in geomaterials, its magnitude will generally vary between the two formations. Different thermal expansion coefficients between the storage formation and the caprock lead to differential expansion of the rock, building up shear stress in the interface between the two layers. When the thermal expansion coefficient of the caprock is greater than that of the storage formation, deviatoric plastic strain may occur in the lower portion of the caprock as a result of cooling (Vilarrasa and Laloui, 2016). Nonetheless, regardless of the stress regime and the relative values of the thermal expansion coefficient between the storage formation and the caprock, the overall sealing capacity of the caprock is not compromised because only the lower portion of the caprock is affected by cooling and the subsequent stress changes.

## 6.  SHEAR SLIP STRESS TRANSFER

Shear slip of faults induce static stress transfer, decreasing stability in some regions, where seismicity rate increases, and increasing stability in others, the so called stress shadows, where seismicity rate decreases or is even suppressed (Harris and Simpson, 1998). Static stress transfer resulting from induced earthquakes has been found to be relevant for explaining post-injection events in EGS stimulations (Schoenball et al., 2012; De Simone et al., 2017). The stress transfer causes rotation of the stress tensor, changing the orientation of the faults that are critically oriented to undergo shear failure. Such change in the orientation of the faults that rupture during water injection and after shut-in was observed at the EGS Basel Deep Heat Mining Project (Deichmann et al., 2014).

Shear slip does not need to be seismic in order to induce stress transfer. Actually, aseismic slip has been reported to indirectly induce seismicity in non-pressurized fault patches (Cappa et al., 2019). The capacity of injection-induced aseismic slip for bringing to failure

zones of faults that are not pressurized has been measured in decameter scale rock laboratories (Guglielmi et al., 2015; Duboeuf et al., 2017). The magnitude of the induced microseismicity in these field experiments is small, in the order of -3.5 (Duboeuf et al., 2017). However, magnitudes may become large in industrial operations if aseismic slip

stresses faults below the injection formation. For example, induced earthquakes with magnitude up to 5 were triggered close to a geothermal plant at Brawley, California, USA (Wei et al., 2015). The accumulated aseismic slip inducing these earthquakes was estimated to be of some 60 cm, nucleating the earthquakes 5 km below the injection formation.

Both seismic and aseismic slip induce stress transfer that affects fracture and fault stability and may induce (micro)seismicity. This effect has been widely studied in natural seismicity, but has received relatively little attention in induced seismicity. Nonetheless, recent studies show that it is a non-negligible effect, which is relevant in post-injection seismicity and for explaining induced events in non-pressurized regions (De Simone et

al., 2017a; Cappa et al., 2019). Thus, even though microseismicity induced by shear slip stress transfer has not been observed to date at geologic carbon storage sites, it should be considered as a potential triggering mechanism.

## 7.   GEOCHEMICAL EFFECTS ON GEOMECHANICAL PROPERTIES

The dissolution of $CO_2$ into the resident brine forms an acidic solution that has the potential of dissolving minerals, which in turn may lead to subsequent precipitation of other minerals (Zhang et al., 2009). The fastest geochemical reactions occur in carbonate rocks and in rocks with carbonate-rich cement (Vilarrasa et al., 2019). Carbonate minerals dissolve when they interact with the acidic $CO_2$-rich brine, leading to porosity and

permeability increase (Alam et al., 2014). The porosity increase leads to a reduction in rock stiffness and strength, which has been measured in the laboratory to be in the order of 20-30% (Bemer and Lombard, 2010; Vialle and Vanorio, 2011; Vanorio et al., 2011; Kim et al., 2018). The measured changes become smaller for increasing confining pressure (Vanorio et al., 2011) because the higher the confinement, the lower the porosity and the available reactive surface and, thus, the reaction rate. The reduction in rock stiffness affects the strain and stress induced by $CO_2$ injection and the reduction in strength may cause failure of initially stable fractures and faults (recall Figure 2), leading to induced microseismicity. Thus, the changes in geomechanical properties of rocks (especially carbonate-rich rocks) as a result of $CO_2$-brine-rock geochemical interactions should be evaluated in the laboratory in order to properly assess the induced microseismicity potential.

Caprocks are also affected to some extent by geochemical reactions. Carbonate and feldspar minerals dissolve in shale, leading to precipitation of other carbonate minerals (Yu et al., 2012). But the overall response of caprocks depend on the rock type. While certain caprocks undergo permeability increase due to interaction with $CO_2$ (Olabode and Radonjic, 2014), others present a self-sealing response to $CO_2$ flow due to porosity decrease (Espinoza and Santamarina, 2012) or fracture clogging (Noiriel et al., 2007). Nevertheless, $CO_2$ is only expected to penetrate a short distance, if any, into the caprock because of its high entry pressure, which prevents upwards $CO_2$ flow (Busch et al., 2008).

For other types of host rocks, laboratory studies have shown that geochemically-induced changes in the geomechanical properties is in general minor (Rohmer et al., 2016). This minor effect has also been observed in fault gouges that have been exposed to acidic conditions for a long period in natural $CO_2$ reservoirs (Bakker et al., 2016). In summary, there is no evidence to expect significant alteration of geomechanical properties induced

by geochemical reactions in general, but (1) the issue should not be abandoned and (2) it should receive especial attention and site-specific studies in carbonate-rich rocks.

## 8. FAULT STABILITY

Faults are present at all scales and have been observed to play a role in $CO_2$ storage projects (e.g., Vidal-Gilbert et al., 2010; Rutqvist, 2012; Castelletto et al., 2013a). To name a few, (i) a fault or fractured rock zone opened as a result of pressure increase at In Salah, Algeria, leading to a double-lobe pattern of uplift on the ground (Vasco et al., 2010; Rinaldi and Rutqvist, 2013); (ii) the storage formation at Snøhvit, Norway, was surrounded by low permeability faults, which limited its storage capacity (Hansen et al., 2013; Chiaramonte et al., 2015); (iii) the Spanish pilot test site at Hontomín contained several minor faults within a few hundreds of meters from the injection well (Alcalde et al., 2013; 2014); and (iv) the pilot test site at Heletz, Israel, is placed in an anticline crossed by two faults, confining the storage formation to be a few hundreds of meters wide (Figueiredo et al., 2015). The nature of these faults, i.e., flow barriers or conduits (Caine et al., 1996), controls the stress changes occurring around the fault and thus, fault stability (Vilarrasa et al., 2016). Low permeability faults may lead to the premature closure of storage sites because of pressure limitations on the storage capacity of the formation (Szulczewski et al., 2012). Actually, if multiple low permeability faults are present and intersecting each other, they will lead to a compartmentalized reservoir (Castelletto et al., 2013b). In such cases, pressure would increase linearly with time (Zhou et al., 2008; Mathias et al., 2011b), increasing injection costs and eventually leading to fault reactivation, and thus induced seismicity, if injection is maintained (Cappa and Rutqvist, 2011a; Pereira et al., 2014; Rutqvist et al., 2016).

Changes in fault permeability due to its reactivation depend on the type of material. Fault reactivation may enhance fault permeability in hard rocks due to dilatancy by one to two orders of magnitude (Cappa and Rutqvist, 2011b; Guglielmi et al., 2015). This permeability increase raises the question of whether fault reactivation may lead to $CO_2$ leakage. Such assessment must be made site specifically taking into account the hydro-mechanical properties of the rock and faults. Nonetheless, in general, faults crossing sequences of reservoirs and caprocks maintain a low-permeability, at least, in the sections that cross caprocks as a result of the high clay content of the fault (Takahashi, 2003; Egholm et al., 2008). But more importantly, the $CO_2$ entry pressure of the fault remains high in the caprock sections (Vilarrasa and Makhnenko, 2017), hindering upwards $CO_2$ leakage, as observed in numerical simulations that incorporate fault heterogeneity (Rinaldi et al., 2014). Additionally, the stress state of the upper crust, which is characterized by a critically stressed crystalline basement overlaid by generally non-critically stressed sedimentary rock (recall Section 3), favors nucleation of the largest seismic events in the crystalline basement rather than in the sedimentary rock where $CO_2$ is stored. This hypocenter distribution has been observed in central US as a result of wastewater injection in the basal aquifer, which is consistent with permeability enhancement below the storage formation but not in the caprock and above, which limits the risk of $CO_2$ leakage (Verdon, 2014).

Apart from $CO_2$ leakage, the magnitude of potential induced earthquakes is a concern because of the damage and fear that they could generate. The magnitude of earthquakes, *M*, is proportional to the rock shear modulus, the rupture area and the mean shear slip (Stekettee, 1958). Thus, the magnitude is controlled by the pressurized area of the fault. In this way, the orientation of the injection well affects the magnitude of potential induced seismicity because wells that are parallel to strata pressurize a larger area than vertical

wells, but take a longer time to exceed the critical pressure at the fault (Rinaldi et al., 2015). The magnitude of induced seismic events is also controlled by the brittleness of the fault. While brittle faults with a slip-weakening behavior can induce large earthquakes ($M$>4) (Rutqvist et al., 2016), ductile faults give rise to progressive ruptures in which shear slip progressively accumulates, giving rise to aseismic slip or a swarm-like seismic activity (Vilarrasa et al., 2017b).

Another aspect that controls fault stability as a result of fluid injection is fault offset. Figure 7a represents a typical scenario that can be encountered in a normal faulting stress regime setting, i.e., a steep fault in which the hanging wall has slid downwards with respect to the footwall. The fault is considered to have an offset equal to half of the storage formation thickness and consists of a low permeability core ($10^{-19}$ m$^2$) and damage zones on the core sides. Properties of the damage zone depend on the material it is in contact with, becoming more permeable and less stiff than the intact rock as a result of fracturing (Table A2). Thus, the damage zone is of high permeability next to the storage formation, but of relatively low-permeability and high entry pressure next to the caprock and base rock. The caprock and base rock are more deformable than the storage formation (Table A3). The model is plane strain, with a constant vertical stress equal to 29.3 MPa acting on the top boundary and no displacement perpendicular to the other boundaries. The top of the storage formation in the hanging wall is placed at 1.5 km depth. $CO_2$ is injected at a constant mass flow rate of $2 \cdot 10^{-3}$ kg/s/m in the hanging wall, 1 km away from the fault, which leads to the pressurization of the storage formation.

Pressure the hanging wall of the storage formation, where $CO_2$ is being injected, increases by up to 10 MPa after 1 year of injection (Figure 7b). The low permeability fault core acts as a flow barrier, causing a rapid reservoir pressurization. This pressure increase expands the storage formation, pushing the fault towards the right-hand side.

While overpressurization is quite uniform across the storage formation, the resistance to displacement on the other side of the fault depends on the stiffness of the rock. Since the storage formation is stiffer than the base rock, it absorbs larger stresses. As a result, the induced horizontal stresses in the in-plane direction are high where the storage formation is present on both sides of the fault, but it is low where the base rock is on the other side of the fault (Figure 7c).

These stress changes have a direct implication on fault stability. Figure 8 displays the changes in the mobilized friction angle around the fault as a result of $CO_2$ injection. The most destabilized region is the lower half of the pressurized storage formation. Thus, an induced microseismic event would be initiated in that region of the fault, but slip would be arrested below the caprock because fault stability improves within the damage zone of the storage formation on the side that is not pressurized. Thus, large magnitude induced events are unlikely in geological settings comparable to this simulated scenario. This difference in fault stability can be easily appreciated by representing Mohr circles in these zones (see inset in Figure 8). Mohr circles shift to the left, getting close to failure, both at the top and bottom of the storage formation due to overpressure. But, while the deviatoric stress is maintained in the lower portion of the pressurized storage formation because the horizontal stress in the in-plane direction does not increase (see red circle in Figure 8), the size of the Mohr circle decreases in the upper portion of the pressurized storage formation because of the increase in the horizontal stress in the in-plane direction where the storage formation is placed on both sides of the fault (see green circle in Figure 8). This fault stability analysis highlights the fact that the accurate assessment of fault stability changes in geologic carbon storage sites completely depend on proper site characterization.

## 9. CHARACTERIZATION TECHNIQUES

Site characterization has been traditionally considered as an activity that should be performed for project design and, therefore prior to operation. This kind of characterization tests are limited in time and can only characterize a small volume of rock around the injection well (Niemi et al., 2017).The size of the region affected by injection grows with the square root of time and since geologic carbon storage projects are planned to last several decades, full characterization can only be achieved by considering operation as a continuous characterization, which we deem necessary to reduce uncertainty in predictive models of felt seismicity.

To assess whether $CO_2$ injection may induce felt seismicity, it is necessary to characterize the geological media in order to build a model of the site. The conceptual model should include the geological layers (at least the caprock, potential secondary caprocks, the storage formation and subjacent layers down to the crystalline basement) and faults. Apart from the geometry, the hydraulic (permeability and porosity), thermal (thermal expansion coefficient, thermal conductivity and heat capacity) and geomechanical (stiffness and strength) properties are required. Additionally, the initial conditions should be determined, i.e., the fluid pressure profile (if pressure is hydrostatic or if there are pressure anomalies), the geothermal gradient, Gutenberg-Richter law and, especially for induced seismicity purposes, the stress state. Determining the magnitude and orientation (and their variability) of the stress tensor is critical, because fault stability depends on the orientation of a given fault with respect to the stress tensor (Morris et al., 1996). Hydraulic, thermal and geomechanical properties of each model layer can be measured in the laboratory from core samples or in the field. While laboratory measurements allow a tight control of test conditions, they usually test only the rock matrix and fail to acknowledge scale effects associated to spatial variability of the above properties and the impact of discontinuities

(e.g., Sanchez-Vila et al., 1996; Ledesma et al., 1996; Zhang et al., 2006; Cai et al., 2007). Thus, interpretation of field tests leads to parameters that are more representative of operation conditions than laboratory experiments.

To obtain estimates representative at the field scale of the hydraulic and geomechanical properties, Vilarrasa et al. (2013c) proposed a hydro-mechanical characterization test for $CO_2$ storage sites (Figure 9). The test consists in injecting water at a high flow rate until microseismic events are induced. Ideally, the same brine from the storage formation should be injected to avoid geochemical reactions around the injection well that may alter rock properties. However, injecting brine would imply having a large facility on surface to store the brine from the storage formation that would have been pumped previously. The test has to be closely monitored with pressure, temperature, deformation and microseismicity monitoring. The hydraulic properties of the storage formation and caprock can be determined from the interpretation of injection as a hydraulic test (Cooper and Jacob, 1946; Hantush, 1956). If heterogeneities are present in the storage formation, their effect is only detectable for a limited period of time (Wheatcraft and Winterberg, 1985; Butler and Liu, 1993). For this reason, it is extremely important to continuously measure pore pressure changes during injection. As for the geomechanical properties of the storage formation and caprock, they can be derived from the interpretation of the vertical displacement at the top of the storage formation and the caprock. Additionally, measuring the pressure evolution in the caprock, which undergoes a pressure drop in response of the pressure buildup in the storage formation (Hsieh, 1996), also gives information on the geomechanical properties. The magnitude of this reverse-water level fluctuation is inversely proportional to the storage formation stiffness (Vilarrasa et al., 2013c). Injection should be maintained until microseismic events are induced in the caprock, which gives an initial estimate of the maximum sustainable injection pressure

that should not be exceeded during $CO_2$ injection to avoid compromising the caprock sealing capacity. This test is valuable to characterize storage sites in a pre-operation stage, but it should be complemented by a continuous site characterization during operation to characterize geological features present in the far field and reduce subsurface uncertainty.

An example of a continuous characterization technique that permits detecting and locating low permeability faults is that proposed by Vilarrasa et al. (2017c). The idea is to use diagnostic plots, i.e., plots that include the fluid pressure evolution together with the derivative of the fluid pressure with respect to the logarithm of time (Bourdet et al., 1983; Renard et al., 2009), to detect faults significantly before (in the order of days) than

if only fluid pressure evolution interpretation would be used (Figure 10a). This early identification of faults should permit decision makers to perform pressure management if necessary to mitigate future induced seismicity. This methodology only detects faults that are at least three orders of magnitude less permeable than the storage formation. However, this should not be a problem in terms of induced seismicity because faults that do not act

as a flow barrier induce relatively small changes in fault stability (Vilarrasa et al., 2016). Low permeability faults generate an additional pressure increase that differs from the expected pressure evolution in an aquifer that would not contain that fault. Thus, by comparing the measured pressure evolution, and its derivative with respect to the logarithm of time, with the predicted one, low permeability faults can be detected. This

additional pressurization also affects the $CO_2$ dynamics because $CO_2$ is pushed away from the direction of the fault, leading to an asymmetric $CO_2$ plume (Figure 10b). Such asymmetry could be detected at monitoring wells, suggesting the presence of a low permeability fault, but it could also be due to reservoir heterogeneity (Chen et al., 2014). Once a fault is detected and located from the interpretation of pressure evolution (Figures

10c and 10d), it should be incorporated into the conceptual model of the site. Additional

characterization techniques may be necessary to obtain a precise information on the detected faults. Then, field measurements should be compared with the updated conceptual model, which will permit identifying and locating new faults (Figure 10c) from the determination of the divergence time and the use of type curves (Figure 10d).

These characterization techniques entail a number of challenges. To begin with, the drilling of a network of monitoring wells is not yet common practice. Additionally, monitoring techniques also present challenges. Pressure is usually measured at the well-head, but calculating the bottom-hole pressure from the well-head pressure is not straightforward given the non-linearities of the injected fluid, especially for $CO_2$ injection

(e.g., Lu and Connell, 2014). Unfortunately, pressure measurements in wells different than the injection well are almost inexistent. Temperature measurements receive even less attention because thermal effects are usually neglected. As for deformation measurements, ground surface can be measured with InSAR data, but for characterization tests that last a few days, the deformation of the ground may not be detectable given the

great depths of suitable storage formations. Thus, deformation should be measured at depth within the boreholes. These measurements pose the question of whether the measured deformation refers to that of the rock or to that of the well. Since the casing of wells is stiffer than rock, the rock may deform more than the well and sliding could even occur between the rock and the cement surrounding the well casing, making accurate

measurements difficult. Fiber optics may solve part of these monitoring challenges, but the way how this monitoring should be performed is still not crystal clear for the moment. As far as microseismicity monitoring is concerned, arrays of geophones should be placed at depth. Otherwise, the signal-to-noise ratio is too high, which complicates detecting microseismic events. Additionally, multi-sensor arrays with a wide aperture coverage are

necessary to accurately locate the events. Despite the existing challenges, such continuous

characterization techniques are needed in order to minimize the risk of inducing seismicity in geologic carbon storage projects.

## 10. MINIMIZING THE RISK OF INDUCING FELT SEISMICITY

The issues discussed in the previous sections make it apparent that it is possible to effectively minimize the risk of inducing earthquakes that are sufficiently large to be felt on the ground surface and may damage structures. We propose here a workflow consisting of the following steps:

1) performing a detailed initial site characterization, with especial emphasis on the geological formations relevant to the site (at least of the storage formation, the caprock, base rock and faults), including the determination of:

- geomechanical properties (Young's modulus, Poisson ratio, cohesion and friction angle);

- hydraulic properties (permeability and porosity);

- thermal properties (thermal expansion coefficient, thermal conductivity and heat capacity);

- the seismic velocities $v_p$ and $v_s$ from the surface to the crystalline basement. An accurate determination of these velocities is important not only for proper interpretation of geophysics, but also to locate the hypocenters of the induced seismicity with precision;

- the baseline of natural seismicity to establish the initial $a$ and $b$ values of the Gutenberg-Richter law in order to distinguish induced from natural seismicity;

- the initial pressure, temperature and stresses profiles with depth from the surface to the crystalline basement. The determination of the stress state is particularly

important to perform a fault stability analysis of the identified faults and determining the strike and dip of critically oriented faults;

- characteristics of geological formations and faults and their location and orientation through 3D seismic data;

2) putting in place proper monitoring for performing continuous characterization, including:

- an array of geophones at depth to measure and locate induced microseismicity;

- a network of geophones on surface or in shallow wells with adequate spatial distribution, covering the whole footprint of the storage site to accurately locate induced seismicity. Induced events should be located in quasi-real time, together with their focal mechanisms to detect potentially unidentified faults that may induce large earthquakes. Inversion of the stress tensor is also important to detect possible local rotations of the stress tensor (Martinez-Garzon et al., 2013, 2014), which could be induced by pressure increase, cooling and/or shear slip stress transfer (De Simone et al., 2017a). This seismic continuous characterization is particularly important when $CO_2$ is injected in the basal aquifer (Verdon, 2014; Will et al., 2016);

- monitoring wells measuring pressure, temperature and $CO_2$ saturation in the storage formation, caprock and secondary aquifer above the storage formation. Monitoring in secondary aquifers is useful for detecting brine and $CO_2$ leakage (e.g., Chabora and Benson, 2009; Zeidouni et al., 2014). Pressure measurements are necessary for continuous characterization techniques as the one described in Section 9;

3) carrying out pressure management:

- storage alternatives to the conventional concept of storing $CO_2$ in deep saline aquifers may be used to have a better control on pressure increase. For example, injection of $CO_2$ dissolved into brine is achieved by creating dipoles of wells in which brine is extracted from the storage formation and reinjected together with $CO_2$ in the same formation (Burton and Bryant, 2009; Jain and Bryant, 2011; Pool et al., 2013). The dipoles of wells limit pressure increase and allow to have a better control on it. Similarly, geothermal energy production using $CO_2$ as a working fluid permits lowering pressure increase and additionally extract geothermal energy (Randolph and Saar, 2011). Despite the promising potential of this technology, the only pilot site that has tried using $CO_2$ as a working fluid yielded a low performance because the thermosyphon that should permit circulating $CO_2$ with a negligible energy consumption did not develop properly (Freifeld et al., 2016). Nevertheless, future research should enable a successful deployment of this technology;

- in any case, predictive models of induced seismicity that consider coupled thermo-hydro-mechanical (THM) processes should be applied to identify the injection scenario that minimizes future induced seismicity. These predictive models should be based on the THM monitoring and continuous characterization. The continuous characterization will permit updating the fault stability analysis by incorporating newly detected faults (recall Figure 10). The range (taking into account the uncertainty on faults properties) of pressure increase that makes faults become critically stressed for shear failure can be determined from the initial stress state, the strike and dip of faults, and the stress changes induced by $CO_2$ injection. Pressure management should be applied to avoid exceeding hazardous levels of pressure increase around faults. To limit

pressure, the injection rate may need to be lowered or pressure may need to be released in the vicinity of critically oriented faults (Birkholzer et al., 2012).

## 11. CONCLUSIONS

Geologic carbon storage can successfully store gigaton scale of $CO_2$ at a low level of induced seismicity provided that proper site characterization, monitoring and pressure management are performed. There are several factors of geologic carbon storage that favor a low induced seismicity risk. First, sedimentary formations where $CO_2$ is planned to be stored are, in general, not critically stressed, which permits generating a certain pressure increase without reaching shear failure conditions. Special care should be taken if $CO_2$ is injected in the basal aquifer, because the crystalline basement is generally critically stressed and may contain unidentified faults that are critically oriented for shear slip. Additionally, $CO_2$ pressure evolution is relatively easy to control because pressure stabilizes after an initial sharp pressure increase, becoming practically constant afterwards. Despite this favorable pressure evolution, if low permeability faults are present, an additional pressure increase may cause large stress changes around the fault, leading to its reactivation. To prevent this situation, a detailed site characterization, both before the start of operation of projects and continuously during the whole operational stage, monitoring and pressure management should permit minimizing the risk of inducing large (felt) earthquakes.

**APPENDIX**

All the presented numerical simulations are performed with the fully coupled finite element code CODE_BRIGHT (Olivella et al., 1994; 1996), which solves non-isothermal two-phase flow in deformable porous media.

5    Table A1. Material properties used in the model of cold $CO_2$ injection shown in Figures 5 and 6

| Property | Reservoir | Caprock and baserock |
|---|---|---|
| Permeability (m$^2$) | $10^{-13}$ | $10^{-18}$ |
| Relative water permeability (-) | $S_l^3$ | $S_l^6$ |
| Relative $CO_2$ permeability (-) | $(1-S_l)^3$ | $(1-S_l)^6$ |
| $CO_2$ entry pressure (MPa) | 0.02 | 0.6 |
| van Genuchten shape parameter (-) | 0.8 | 0.5 |
| Porosity (-) | 0.15 | 0.01 |
| Young's modulus (GPa) | 10.5 | 5.0 |
| Poisson ratio (-) | 0.3 | 0.3 |
| Cohesion (MPa) | 0.01 | 0.01 |
| Friction angle (-) | 30.0 | 27.7 |
| Thermal conductivity (W/m/K) | 2.4 | 1.5 |
| Solid specific heat capacity (J/kg/K) | 874 | 874 |
| Linear thermal expansion coefficient (ºC$^{-1}$) | $10^{-5}$ | $10^{-5}$ |

$S_l$ is the liquid saturation degree

Table A2. Properties of the materials forming the fault of the model shown in Figures 7 and 8

| Property | Fault core | Damage zone reservoirs | Damage zone confinement layers | Damage zone basement |
|---|---|---|---|---|
| Permeability (m$^2$) | $1 \cdot 10^{-19}$ | $2 \cdot 10^{-13}$ | $1.5 \cdot 10^{-19}$ | $1 \cdot 10^{-16}$ |
| Relative water permeability (-) | $S_l^6$ | $S_l^3$ | $S_l^6$ | $S_l^4$ |
| Relative $CO_2$ permeability (-) | $(1-S_l)^6$ | $(1-S_l)^3$ | $(1-S_l)^6$ | $(1-S_l)^4$ |
| $CO_2$ entry pressure (MPa) | 4.0 | 0.02 | 5.0 | 1.0 |
| van Genuchten shape parameter (-) | 0.3 | 0.8 | 0.3 | 0.5 |
| Porosity (-) | 0. 10 | 0.25 | 0.09 | 0.07 |
| Young's modulus (GPa) | 1.0 | 7.0 | 1.4 | 42.0 |
| Poisson ratio (-) | 0.30 | 0.35 | 0.42 | 0.30 |

$S_l$ is the liquid saturation degree

Table A3. Material properties of the intact rock types included in the model shown in Figures 7 and 8

| Property | Storage formation | Caprock | Base rock | Upper aquifer | Crystalline basement |
|---|---|---|---|---|---|
| Permeability (m$^2$) | $4 \cdot 10^{-14}$ | $8 \cdot 10^{-20}$ | $5 \cdot 10^{-20}$ | $1 \cdot 10^{-14}$ | $4 \cdot 10^{-20}$ |
| Relative water permeability (-) | $S_l^3$ | $S_l^6$ | $S_l^6$ | $S_l^3$ | $S_l^6$ |
| Relative $CO_2$ permeability (-) | $(1-S_l)^3$ | $(1-S_l)^6$ | $(1-S_l)^6$ | $(1-S_l)^3$ | $(1-S_l)^6$ |
| $CO_2$ entry pressure (MPa) | 0.02 | 10.0 | 10.0 | 0.20 | 12.0 |

| | | | | | |
|---|---|---|---|---|---|
| van Genuchten shape parameter (-) | 0.8 | 0.3 | 0.3 | 0.8 | 0.3 |
| Porosity (-) | 0.23 | 0.05 | 0.05 | 0.13 | 0.01 |
| Young's modulus (GPa) | 14.0 | 2.8 | 3.0 | 28.0 | 84.0 |
| Poisson ratio (-) | 0.31 | 0.40 | 0.39 | 0.21 | 0.18 |

$S_l$ is the liquid saturation degree

**ACKNOWLEDGEMENTS**

VV acknowledges funding from the European Research Council (ERC) under the European Union's Horizon 2020 research and innovation programme (grant agreement No. 801809). JR acknowledges funding by the Assistant Secretary for Fossil Energy, National Energy Technology Laboratory, National Risk Assessment Partnership of the U.S. Department of Energy to the Lawrence Berkeley National Laboratory under Contract No. DEAC02-05CH11231.

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

**TABLES**

Table 1. Stress state (maximum principal stress, $\sigma_1$, intermediate principal stress, $\sigma_2$, minimum principal stress, $\sigma_3$, and pore pressure, $P$) and mobilized friction coefficient ($\mu_{mob}$) at several $CO_2$ injection sites

| Site | Depth (m) | $\sigma_1$ (MPa) | $\sigma_2$ (MPa) | $\sigma_3$ (MPa) | $P$ (MPa) | $\mu_{mob}$ (-) |
|---|---|---|---|---|---|---|
| In Salah, Algeria[1] | 1800 | 49.9 | 44.5 | 30.8 | 18.2 | 0.48 |
| Weyburn, Canada[2] | 1450 | 34.0 | 26.0 | 22.0 | 14.5 | 0.50 |
| Otway, Australia[3] | 2000 | 58.0 | 44.0 | 31.0 | 8.6 | 0.41 |
| Snøhvit, Norway[4] | 2683 | 65.0 | 60.6 | 43.0 | 29.0 | 0.49 |
| Tomakomai, Japan[5] | 2352 | 53.8 | | 43.8 | 33.7 | 0.35 |
| St. Lawrence Lowland, Canada[6] | 1200 | 48.0 | 30.7 | 24.6 | 11.8 | 0.54 |
| Decatur, Illinois[7] | 2130 | 98.0 | | 50.6 | 21.9 | 0.51 |
| Pohang, Korea[8] | 775 | 18.2 | 15.1 | 13.8 | 7.6 | 0.27 |

References: [1] Morris et al. (2011), [2] White and Johnson (2009), [3] Nelson et al. (2006); Vidal-Gilbert et al. (2010), [4] Chiaramonte et al. (2013), [5] Kano et al. (2013), [6] Konstantinovskaya et al. (2012), [7] Bauer et al. (2016), [8] Lee et al. (2017)

**FIGURES**

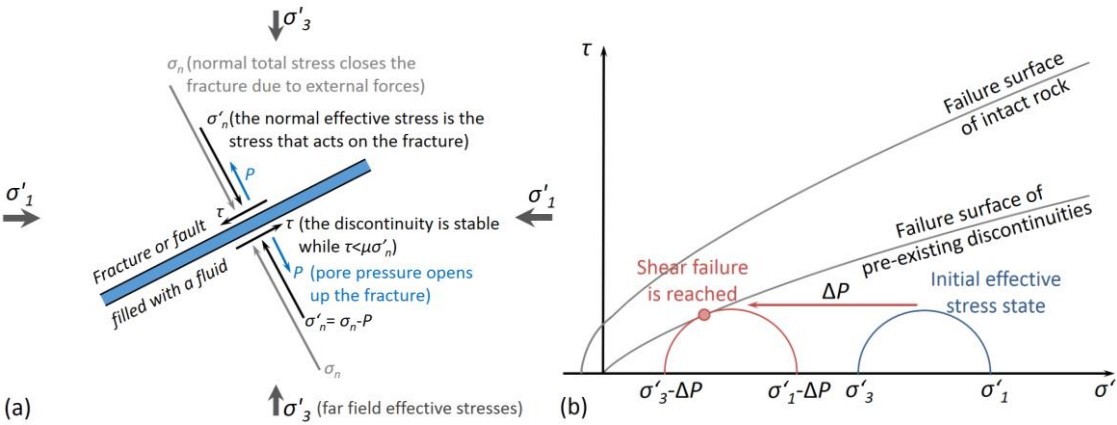

Figure 1: (a) Initial stress state of a fracture or fault of arbitrary orientation with respect to the far field effective stress and (b) Mohr circles showing how the reduction in effective stresses as a result of pressure buildup, $\Delta P$, may induce shear failure in pre-existing discontinuities, i.e., fractures or faults. $\sigma'_1$ and $\sigma'_3$ are the maximum and minimum principal effective stresses, respectively, $\tau$ is tangential stress, $\sigma'_n$ is normal effective stress to the fracture or fault, and $\mu$ is the friction coefficient. The failure surface has been plotted considering non-linear failure criterion (Barton, 1976).

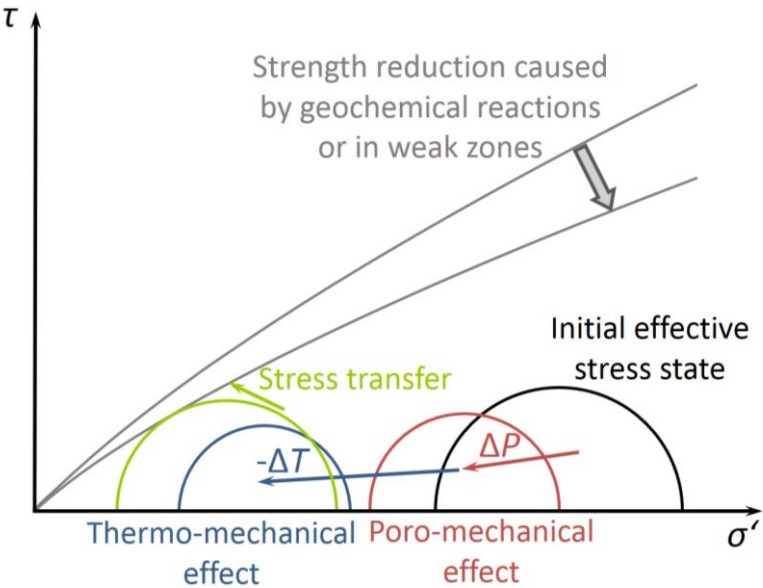

Figure 2: Schematic representation of several coupled effects on fracture/fault stability. Pressure buildup, $\Delta P$, decreases the effective stresses and may cause poro-mechanical stresses that change the size of the Mohr circle; cooling, $-\Delta T$, induces thermal stress reduction; seismic and aseismic shear slip and interactions between geological layers with different rock properties produce total stress changes; and geochemical reactions may alter the strength of fractures and/or faults.

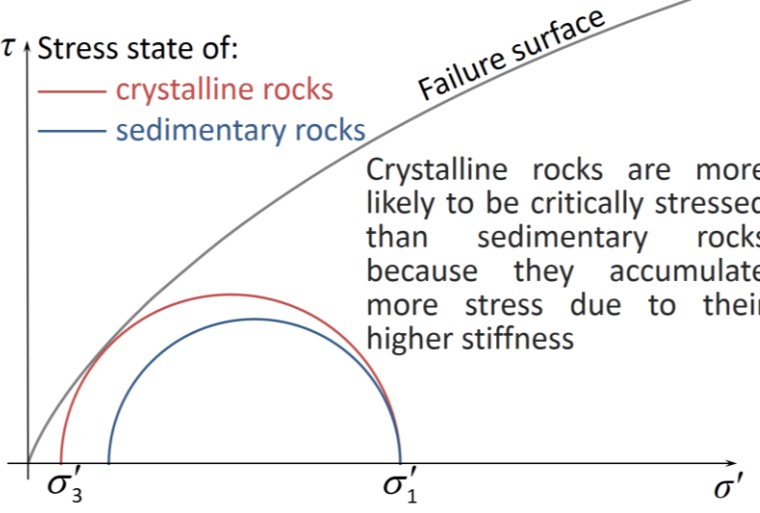

Figure 3: Schematic representation of the stress state of crystalline and sedimentary rocks, showing that sedimentary rocks, which are the rocks where $CO_2$ will be stored, are generally less stressed than the crystalline basement.

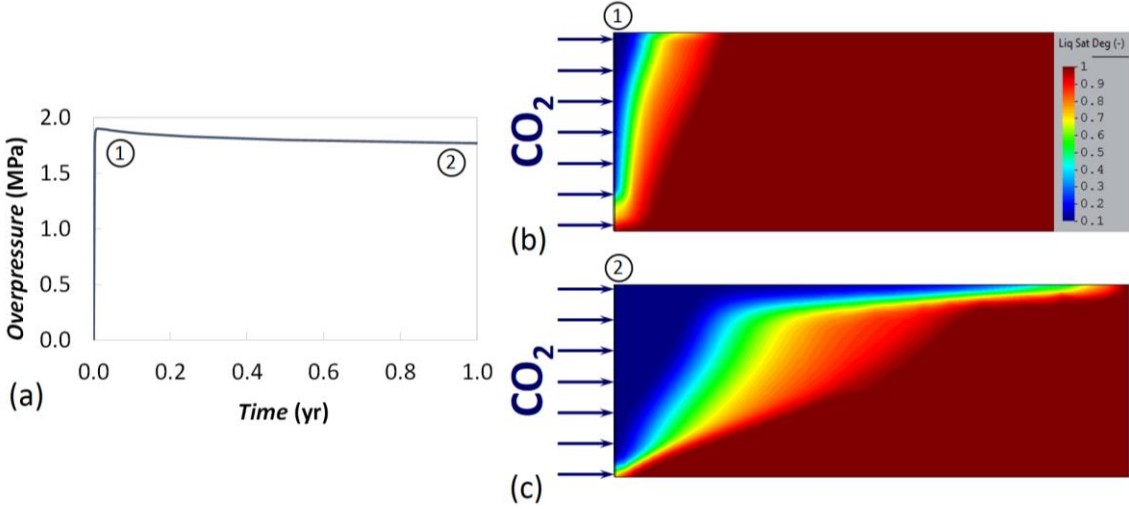

Figure 4: (a) $CO_2$ pressure evolution when injecting 1 Mt/yr of $CO_2$ through a vertical well in a 100-m thick aquifer with an intrinsic permeability of $10^{-13}$ m$^2$ and a radius of 100 km, (b) the $CO_2$ plume shape at the beginning of injection, coinciding with the peak in injection pressure (see number 1 in (a)), and (c) the $CO_2$ plume once gravity override dominates and the capillary fringe has been developed, leading to a slight pressure drop (see number 2 in (a)). The color bar displaying the liquid saturation degree in (b) applies for both (b) and (c).

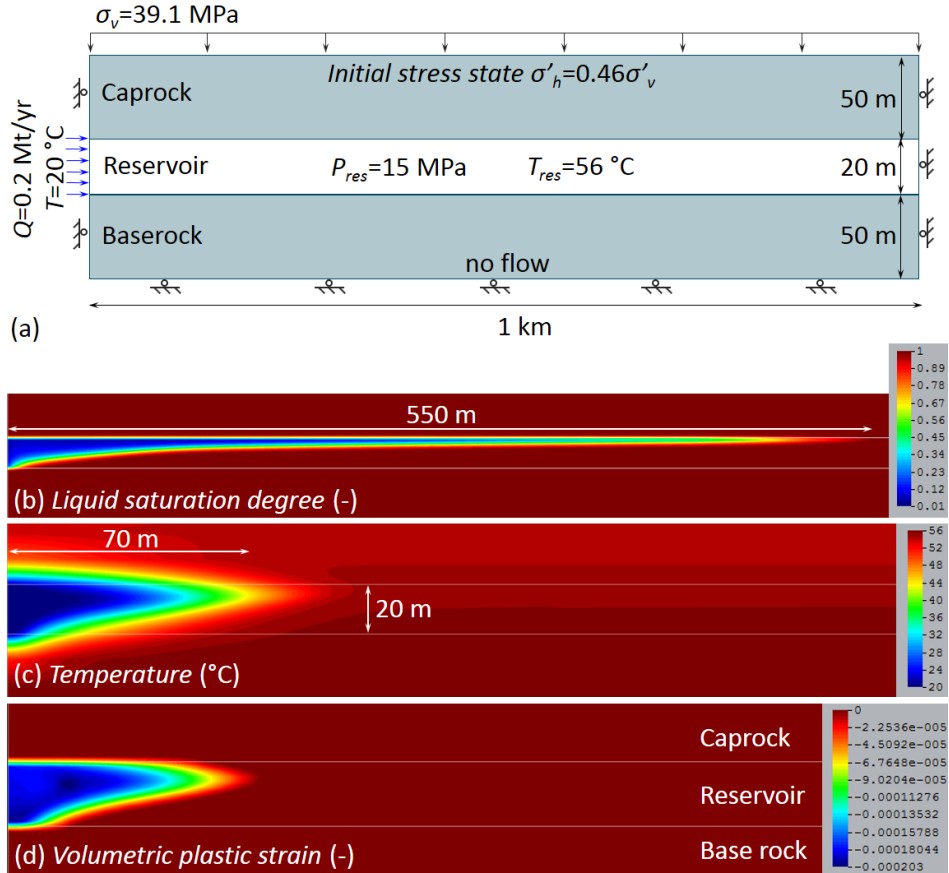

Figure 5: (a) Model setup, (b) liquid saturation degree, (c) temperature distribution and (d) volumetric plastic strain after 2 years of injecting 0.2 Mt/yr of $CO_2$ at 20 ℃ through a vertical well. While (c) and (d) are plotted at the same scale, (b) is plotted at a smaller scale.

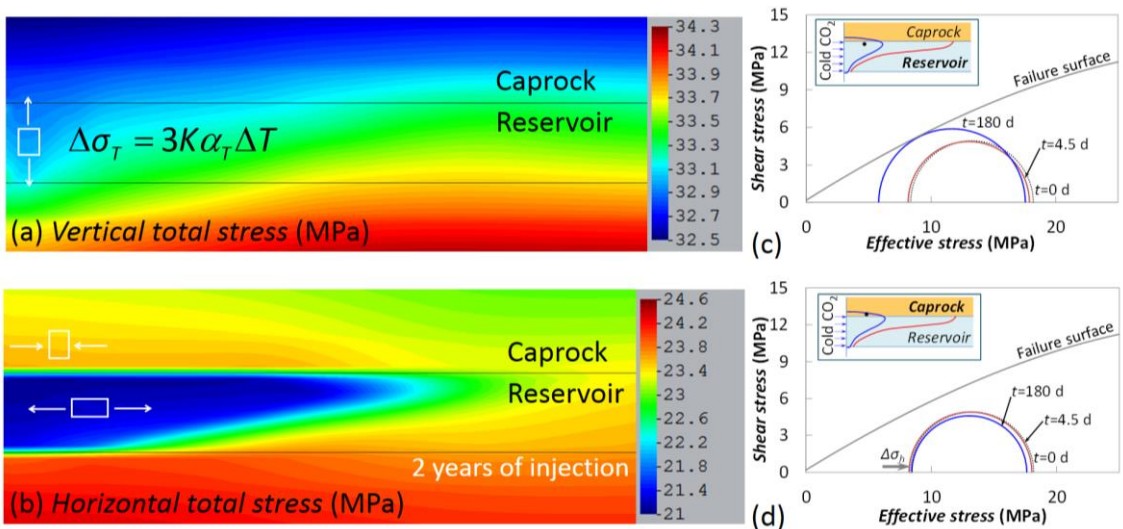

Figure 6: Total stresses in the (a) vertical and (b) horizontal direction after 2 years of injecting 0.2 Mt/yr of $CO_2$ at 20 ℃ through a vertical well, indicating the sign of the induced stresses. Thermal stresses, $\Delta\sigma_T$, are proportional to the bulk modulus, $K$, the thermal expansion coefficient, $\alpha_T$, and the temperature difference, $\Delta T$. The changes in the Mohr circles at a point placed 25 m away from the injection well in (c) the reservoir (2 m below the reservoir-caprock interface) and (d) the caprock (2 m above the reservoir-caprock interface) are also represented.

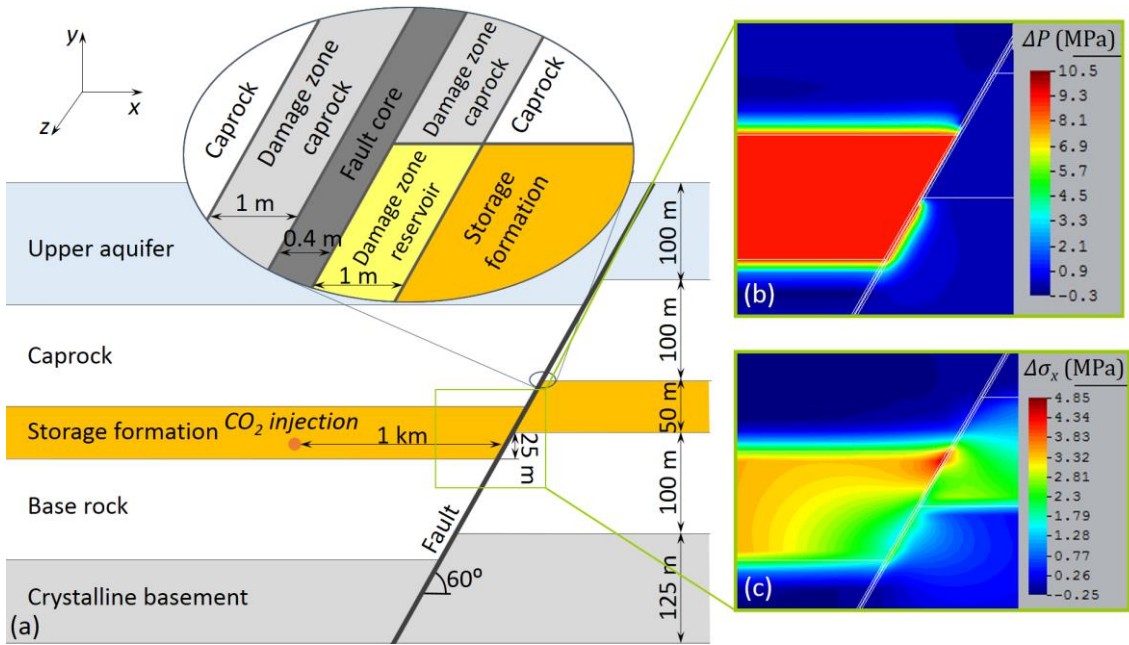

Figure 7: (a) Geological setting in a normal faulting stress regime (plane strain model), including a low permeability fault that leads to (b) reservoir pressurization, $\Delta P$, and (c) horizontal total stress changes in the in-plane direction, $\Delta \sigma_x$, when $CO_2$ is injected in the hanging wall at a rate of $2 \cdot 10^{-3}$ kg/s/m for 1 year.

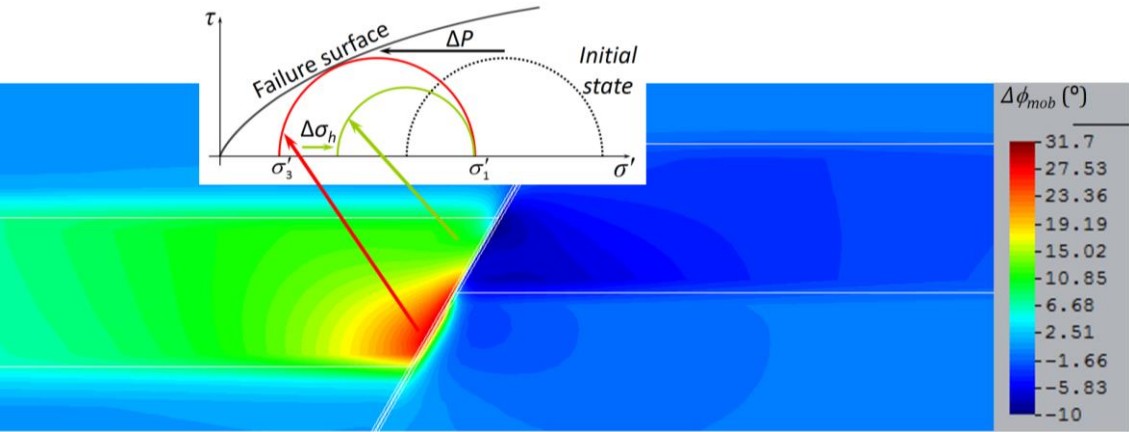

Figure 8: Distribution of stability changes induced by the pressure and stress changes shown in Figure 7, measured in terms of the mobilized friction angle changes, $\Delta\phi_{mob}$. The inset shows the Mohr circles before and after reservoir pressurization.

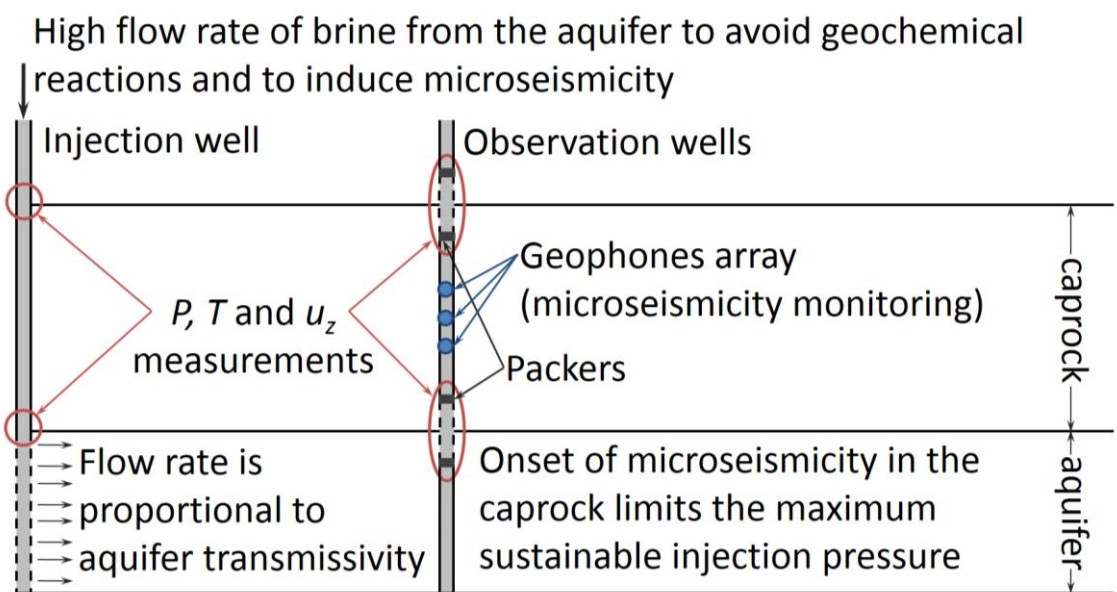

Figure 9: Hydro-mechanical characterization test proposed by Vilarrasa et al. (2013c) to quantify the rock properties at the field scale and obtain an initial estimate of the maximum sustainable injection pressure. *P* refers to pressure, *T* to temperature and $u_z$ to vertical displacement.

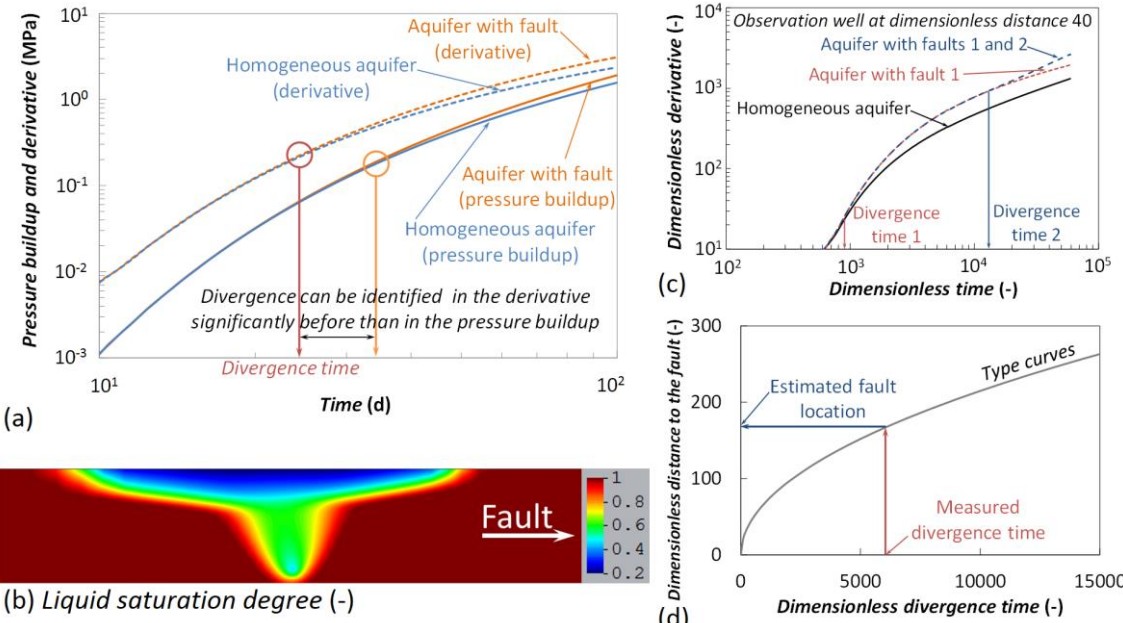

Figure 10: (a) Concept of the continuous characterization technique proposed by Vilarrasa et al. (2017c) to detect and locate low permeability faults using diagnostic plots; (b) asymmetric $CO_2$ plume as a result of the additional pressurization caused by a low-permeability fault, which displaces $CO_2$ towards the opposite direction of the fault; (c) detection of multiple faults by updating the conceptual model of the site and comparing field measurements with predictive simulations; and (d) estimation of the fault location from the measured divergence time in the derivative of the pressure evolution using type curves.