# Peer review of "Induced seismicity in geologic carbon storage"

_Solid Earth, 2018_

## Referee Comment (RC1) · Anonymous Referee #1 · 5 Feb 2019

General Comments

The authors present and review an overview of the issues surrounding induced seismicity in geologic carbon storage. Specifically the authors attempt to show the impacts of 1) stress state, 2 ) pressure evolution, 3) thermal effects, and 4) fault stability on the potential for induced seismicity. They then assess the characterisation required to analyse the above and propose a number of ways to minimise the risk of induced seismicity.

Whilst each of the above are treated suitably I struggle to see the major advances in this paper (above that of the cited papers) as required for a research article. It almost has the feel of a review/commentary paper. This may be enhanced by the lack of clarity on what original research is presented here as opposed to previously published citations (of which more than 130 also makes this feel more like a review). I give examples

below. Specifically, there is no introduction or methods that describe what numerical modelling is actually performed. If there are new results here, they need to be shown more clearly.

Specific Comments

Page 6. Triggering mechanisms. Many alternative mechanisms (other than pore pressure increase) are presented for seismicity triggers but the paper then only goes on to explore a few of these explicitly. For example, heterogeneity and geochemical effects are not discussed further. Thermal effects are considered but no detailed assessment of rock properties and contrast of layers. No further discussion of stress redistribution or aseismic slip. Thus I am left feeling the conclusion that seismicity can be predicted, monitored and managed is undermined by not tackling these in detail.

Page 8. Stress state It is a very large assumption to say sedimentary rocks are not critically stressed. There are clearly many exampled (even cited in this paper, e.g. Blackpool) where sedimentary rocks are critically stressed. The last sentence of this section admits this but it does not appear valid to me to makes this strong assertion/assumption, particularly as displayed in Figure 3.

Page 9 Pressure Buildup Evolution. This may be many, and even incorrect, but is the term "pressure buildup" used correctly here? This phrase, to me, implies the early stage of injection, or the build-up to max sustained pressure. Here it used to describe what i would call 'pressure evolution' over the whole project.

In the discussion of Figure 4, is this new work? How was it modelled? What are the boundary conditions, scales etc etc? Labelling on the figure also needs improved.

Page 10. Here the authors state that pressure dissipation can be accommodated by brine leaking through a fault but not CO2. They need to be explicit as to why this is the case. e.g. in the last sentence of this section this should state there is high entry pressure 'to CO2' specifically and that there is (presumably) a lower entry pressure for

brine.

Page 11. Non Isothermal Effects As with the pressure modelling, is this new work here? For example, on lines 6-10 of page 11 is this new work or results from Jeanne et al? lines 11-13. Is this a general comment or for a specific model/conditions? lines 16-18. Why? Is this because there is only cooling in the reservoir not caprock? line 19. Why especially in normal faulting regimes? line 21 - end of section. Is all this discussion (and figure 6) all based on modelling? As for above, what conditions, modelling approach etc etc if it is new.

Page 14 Fault Stability. line 6-8. Surely depends on the orientation of the strata (if in sed rocks) relative to the well, not that the well is horizontal? line 24-25. What does 'more deformable' mean? Is this a condition set in the model? page 15 line 8. Why is reservoir stiffer? Is this a condition of the model again?

Characterization techniques pg 17 line 1. Stress orientations and magnitudes are pretty hard to measure from core. Can this be changed to 'most properties'. pg 18 line 2. Do we need to be careful here about formation/caprock damage here? How is this different/beneficial to say a XLOT in the caprock? pg 19 line 1. Heterogeneity is the crucial bit here. I'm not sure you can confidently infer the next section (and figure 10) when heterogeneity could easily give the same results.

Minimising Risk. pg 21. line 16 onwards. This section/bulletpoint seems a little out of place here. Sure, co-injection etc. could be used but there are other ways to manage pressure too (from straight water production and disposal to changing injection rate, WAG or not etc etc) and for a section entitled other storage concepts there are lots of other methods (basalt storage e.g.). The link to geothermal energy seems out of place/unnecessary.

Figures 4-8 in particular need more scale bars, description of colours used etc. Fig 5 in particular needs better labelling to show which mohr diagram is for which layer.

[Figure]

[Figure]

---

## Referee Comment (RC2) · Anonymous Referee #2 · 12 Mar 2019

Lacking in new data and contributions, and filled with citations and unsupported generic comments The authors attempt to mitigate undesirable induced seismicity by investigating different mechanisms leading to fracture/fault instability and performing numerical simulations. The authors mention that the main factors causing stress changes in the reservoir are injection-related pressure buildup, in-situ stress state, injected fluid's temperature gradient. The outline of the paper is communicated at the end of Section 1 in page 4. However, there is no clear section on what unique contributions this study is making to improve the state-of-the-art. A general theme of the manuscript is that too many generic, qualitative comments are made without new data or analysis to support those comments. There is an unreasonably large emphasis on citing and reviewing existing papers instead of showing new results. When the simulation results are shown, there are no clear quantitative details of the simulation model: model dimensions, meshing, initial and boundary conditions, well conditions, and hy-

draulic/mechanical properties. This suggests that the manuscript should be submitted as a review article, not Research Article. 1. Figure 1,2,3: They are extremely generic, redundant and partially inaccurate. For example, Figure 2 shows that the effect of temperature change is to only shift the Mohr Circle to left, which is highly imprecise and can be inaccurate depending on the rock type, injection layer geometry (total stress can change), and the magnitude and direction of temperature change. Figure 3 lumps all sedimentary rocks in the world as critically unstressed and assumes that they all fail under linear Mohr Coulomb condition. This is almost unscientific and completely unnecessary. 2. Figure 4: This shows results for a problem that is not even defined. What is the physical model setup, what are the initial and boundary conditions of the coupled flow-mechanics problem, what is the well rate and injection duration? Why do we accept this result as correct? 3. Figure 5: Same as before. Why is this an accepted solution? What is the problem setup? 4. Page 9: "progressively increasing the flow rate at the beginning of injection may avoid the initial peak in pressure buildup" This statement needs to be quantified: how much increase to avoid how much pressure buildup. Otherwise, the idea of "progressively increasing the rate" is a conjecture. 5. Page 1-15: There is too much literature review. Almost 906. Abstract: "We aim at understanding . . . and to develop methodologies . . . through dimensional and numerical analysis." There is now dimensional analysis. In fact, the word "dimensional" appears only once in the abstract. Please remove it from the abstract. 7. Page 14-15: This combines citations with discussion of authors' results. This is very confusing. It is better to move authors' own work into a separate section and not mix with background literature survey. 8. Page 15 line 5: "As a result, the induced horizontal stresses in the in-plane direction are high where the storage formation is present on both sides of the fault, but it is low where the base rock is on the other side of the fault." This is not a result in this manuscript. Either remove it or support it with actual simulation results. 9. Figure 7 and 8: Data used for the simulation must be provided otherwise it is not clear what to expect in the result. What is the contrast in elastic stiffness and hydraulic properties between the damage zone vs. reservoir vs. caprock. All modeling assumptions

used during the simulation must be listed. 10. Page 17-18: This proposes a field test to macroscopically characterize hydraulic, thermal and geomechanical properties without mentioning any challenges related to applicability and operation. Otherwise such a field test will get classified as unrealistic and not useful for $CO_2$ injection. 11. Page 21: "predictive models of induced seismicity that consider coupled THMS processes should be applied" This is much easier said than done. What are these models? The results in this manuscript do not show any coupling to seismicity, which requires solution of the elastodynamic problem in a n-dimensional domain with a (n-1) dimensional fault surface, not a n-dimensional fault zone. This manuscript presents neither an approach nor results from coupling of the four processes T, H, M, S. 12. Page 21: "The continuous characterization will permit updating the fault stability analysis by incorporating newly detected faults." How will the new faults be detected? This is not trivial and not answered in this manuscript. So, please remove this. 13. Figure 6: Color scale can be improved. For example, it is different for the upper and lower figures, yet the maximum value is not visible in the upper figure.

---

## Author Comment (AC2) · 29 Mar 2019

RESPONSE TO THE INTERACTIVE COMMENT OF REFEREE #2

We discuss below the comments made by the reviewers and our responses. To facilitate reading, we indicate the referee's comments with C and our responses with Reply.

General Comments

C: The authors attempt to mitigate undesirable induced seismicity by investigating different mechanisms leading to fracture/fault instability and performing numerical simulations. The authors mention that the main factors causing stress changes in the reservoir are injection-related pressure buildup, in-situ stress state, injected fluid's temperature gradient. The outline of the paper is communicated at the end of Section 1 in page

4. However, there is no clear section on what unique contributions this study is making to improve the state-of-the-art. A general theme of the manuscript is that too many generic, qualitative comments are made without new data or analysis to support those comments. There is an unreasonably large emphasis on citing and reviewing existing papers instead of showing new results. When the simulation results are shown, there are no clear quantitative details of the simulation model: model dimensions, meshing, initial and boundary conditions, well conditions, and hydraulic/mechanical properties. This suggests that the manuscript should be submitted as a review article, not Research Article.

Reply: As we already explained in the response to the interactive comment of referee #1, this is a review article, because as awardee of the Outstanding Early Career Scientists Award for the Division on Energy, Resources and the Environment (ERE) of the EGU, I was invited to publish a paper in one of the EGU journal based on my lecture. Since I presented in my lecture the work that I have done in the last years and that contributed to receive the award, the article type should be changed from research article to review article. We apologize for this mistake when we submitted the manuscript.

Specific Comments

C: Figure 1,2,3: They are extremely generic, redundant and partially inaccurate. For example, Figure 2 shows that the effect of temperature change is to only shift the Mohr Circle to left, which is highly imprecise and can be inaccurate depending on the rock type, injection layer geometry (total stress can change), and the magnitude and direction of temperature change. Figure 3 lumps all sedimentary rocks in the world as critically unstressed and assumes that they all fail under linear Mohr Coulomb condition. This is almost unscientific and completely unnecessary.

Reply: These three Figures are schematic to explain general aspects of induced seismicity. Regarding the shift of the Mohr circle due to temperature change, it is shifted to the left because cooling is expected to occur around CO2 injection wells, and thus, a

total stress reduction will occur. We will add a minus in front of the delta T to indicate that cooling takes place. Additionally, the size of the Mohr circle changes because the changes in the total stresses may be different in the vertical and horizontal directions. Nonetheless, it may be difficult to observe that the two circles (the red and the blue ones) have different sizes, so we will modify the Figure to exaggerate this effect. As for Figure 3, we agree with the referee that not all sedimentary rocks are not critically stressed, as we already state in the figure caption and main text. We also agree with the referee that the failure envelope is not linear for rock. Indeed, we usually use non-linear shear strength in our studies. Since the Figure was schematic, we were just representing a linear failure surface, but we will modify it to show the non-linearity of shear strength. Additionally, we will modify this Figure to indicate that crystalline rock is more likely to be critically stressed than sedimentary rocks because of their higher stiffness, which makes them accumulate more stress. Additionally, to support this statement, we will add a Table showing the stress state at several $CO_2$ storage sites together with the mobilized friction coefficient. The mobilized friction coefficient ranges from 0.35 to 0.54, so in all cases is lower than 0.6, meaning that favourably oriented faults to undergo shear slip are not critically stressed. Of course, knowing the stress state at each site is crucial because the maximum sustainable injection pressure to avoid reactivating faults depends on the initial stress of state. Thus, the injection pressure at the site with a mobilized friction coefficient of 0.54 has to be lower than at the site with a mobilized friction coefficient of 0.35.

C: Figure 4: This shows results for a problem that is not even defined. What is the physical model setup, what are the initial and boundary conditions of the coupled flow-mechanics problem, what is the well rate and injection duration? Why do we accept this result as correct?

Reply: This Figure describes the pressure evolution in a 100-m thick reservoir in which 1 Mt of $CO_2$/yr are injected in an aquifer with permeability of 1e-13 m2 and radius of 100 km. Since the pressure front does not reach the outer boundary during the injection

period shown in the figure, the nature of the boundary does not have any effect on the pressure evolution. The aquifer, which is placed at 1.5 km depth, initially presents hydrostatic pressure. Nevertheless, since we show the pressure changes, the absolute initial pressure is not relevant. We will provide these details on the characteristics of this particular model in the manuscript. Regardless of the particularities of this model, the intention is to describe in a general way $CO_2$ injection pressure evolution, which is significantly different from that of water injection. As explained in the text, the characteristics of this pressure evolution, i.e., the initial sharp increase in $CO_2$ pressure followed by a relatively constant injection pressure, have been observed in the field, in analytical and numerical solutions. Based on this evidence, it can be accepted as correct.

C: Figure 5: Same as before. Why is this an accepted solution? What is the problem setup?

Reply: The results shown in this Figure are from a fully coupled numerical code that solves non-isothermal two-phase flow in deformable porous media (CODE_BRIGHT), which has been benchmarked extensively and is well accepted within the scientific community. As for the problem setup, we will add more details. Nevertheless, the Figure was intended to support the explanations of the processes that occur during cold $CO_2$ injection, without focusing on a specific case.

C: Page 9: "progressively increasing the flow rate at the beginning of injection may avoid the initial peak in pressure buildup" This statement needs to be quantified: how much increase to avoid how much pressure buildup. Otherwise, the idea of "progressively increasing the rate" is a conjecture.

Reply: We will delete this sentence.

C: Page 1-15: There is too much literature review. Almost 90.

Reply: We deem this amount of references appropriate for a review article.

C: Abstract: "We aim at understanding . . . and to develop methodologies . . . through dimensional and numerical analysis." There is now dimensional analysis. In fact, the word "dimensional" appears only once in the abstract. Please remove it from the abstract.

Reply: We will remove the word dimensional in the abstract.

C: Page 14-15: This combines citations with discussion of authors' results. This is very confusing. It is better to move authors' own work into a separate section and not mix with background literature survey.

Reply: In this section, we are providing explanations of the relevant aspects that control fault stability. Even though we have studied this problem extensively, other authors have made relevant contributions to the topic and we believe that it is important to include their contributions in this section as well.

C: Page 15 line 5: "As a result, the induced horizontal stresses in the in-plane direction are high where the storage formation is present on both sides of the fault, but it is low where the base rock is on the other side of the fault." This is not a result in this manuscript. Either remove it or support it with actual simulation results.

Reply: This statement results from the observation of the changes in the horizontal stress in the in-plane direction shown in Figure 7. To clarify this point, we will add a reference to this Figure at the end of the sentence.

C: Figure 7 and 8: Data used for the simulation must be provided otherwise it is not clear what to expect in the result. What is the contrast in elastic stiffness and hydraulic properties between the damage zone vs. reservoir vs. caprock. All modeling assumptions used during the simulation must be listed.

Reply: We will provide this information in the revised version of the manuscript.

C: Page 17-18: This proposes a field test to macroscopically characterize hydraulic, thermal and geomechanical properties without mentioning any challenges related to

applicability and operation. Otherwise such a field test will get classified as unrealistic and not useful for CO2 injection.

Reply: We thank the referee for raising this point, which is certainly of interest and deserves discussion. There are a number of challenges related to this characterization test. To begin with, the drilling of a network of monitoring wells is not a common practice yet. Monitoring techniques also present challenges. Pressure is usually measured at the well-head, but calculating the bottom-hole pressure from the well-head pressure is not straightforward given the non-linearities of the injected fluid, especially for CO2 injection. Unfortunately, pressure measurements in well different than the injection well are almost inexistent. Temperature measurements receive even less attention. As for deformation measurements, ground surface can be measured with InSAR data, but for characterization tests that last a few days, the deformation of the ground may not be detectable given the great depths of storage formations. Thus, deformation should be measured at depth within the boreholes. These measurements pose the question of whether the measured deformation refers to that of the rock or to that of the well. Since the casing of wells is stiffer than rock, the rock may deform more than the well and sliding could even occur between the rock and the cement surrounding the well casing. Fiber optic may solve part of these monitoring challenges, but the way how this monitoring should be performed is still not crystal clear for the moment. As far as microseismicity monitoring is concerned, arrays of geophones are certainly needed to be placed at depth. Otherwise, the signal-to-noise ratio is too high, which complicates detecting microseismic events. Additionally, multi-sensor arrays with a wide aperture coverage are necessary to accurately locate the events. We will include in the manuscript this discussion on the challenges of performing such characterization test.

C: Page 21: "predictive models of induced seismicity that consider coupled THMS processes should be applied" This is much easier said than done. What are these models? The results in this manuscript do not show any coupling to seismicity, which

requires solution of the elastodynamic problem in a n-dimensional domain with a (n-1) dimensional fault surface, not a n-dimensional fault zone. This manuscript presents neither an approach nor results from coupling of the four processes T, H, M, S.

Reply: This is a recommendation we made for future practices based on our previous experience. Given that we do not go into the details of the seismic part, we will replace THMS by THM, which is discussed in the manuscript.

C: Page 21: "The continuous characterization will permit updating the fault stability analysis by incorporating newly detected faults." How will the new faults be detected? This is not trivial and not answered in this manuscript. So, please remove this.

Reply: The continuous characterization refers to the methodology explained in Figure 10. Thus, by applying this methodology, it is possible to detect previously unidentified low-permeable faults and incorporate them in the model of the injection site. We will mention Figure 10 at the end of this sentence to clarify how new faults can be detected.

C: Figure 6: Color scale can be improved. For example, it is different for the upper and lower figures, yet the maximum value is not visible in the upper figure.

Reply: We will improve the color scales of Figure 6 so that the maximum and minimum values are visible.

---

## Author Response (AR1)

Dear Sir,

We would like to begin by thanking you and the referees for all of your efforts with this manuscript. We would like to kindly ask you to change the article type from Research article to Review paper. I apologize for choosing the wrong article type, which has caused some confusion to the referees.

The comments have been quite constructive and we have incorporated most of the suggestions made by the referees. We believe that the explanations provided below will help in clarifying our assumptions. We have also modified the original manuscript to make them clear to all readers. As a result, we feel that this revised version has improved with respect to the original manuscript. Please find our detailed responses to each of the referees' comments below.

Sincerely yours,

Victor Vilarrasa

**RESPONSE TO REFEREES' COMMENTS**

We discuss below the comments made by the referees and our responses. To facilitate reading, we indicate the referee's comments with C and our responses with Reply. We also indicate how we have addressed the comments in the revised manuscript with Authors' changes.

**REFEREE #1**

**General Comments**

C: The authors present and review an overview of the issues surrounding induced seismicity in geologic carbon storage. Specifically the authors attempt to show the impacts of 1) stress state, 2) pressure evolution, 3) thermal effects, and 4) fault stability on the potential for induced seismicity. They then assess the characterisation required to analyse the above and propose a number of ways to minimise the risk of induced seismicity.

Reply: We would like to begin by thanking the referee for looking in detail to this manuscript, as shown by this concise summary.

C: Whilst each of the above are treated suitably I struggle to see the major advances in this paper (above that of the cited papers) as required for a research article. It almost has the feel of a review/commentary paper. This may be enhanced by the lack of clarity on what original research is presented here as opposed to previously published citations (of which more than 130 also makes this feel more like a review). I give examples below. Specifically, there is no introduction or methods that describe what numerical modelling is actually performed. If there are new results here, they need to be shown more clearly.

Reply: We understand the referee's concern and would like to explain the peculiarity of this manuscript. As awardee of the Division Outstanding Early Career Scientists Award for the Division on Energy, Resources and the Environment (ERE) of the EGU, I was invited to publish a paper in one of the EGU journals based on my lecture. In the lecture, I presented the work that I have done in the last years and that contributed to receive the award. This is why the paper is a review and compilation of recent work. To avoid this kind of misunderstanding, we are asking the editor to change the article type from Research article to Review paper.

**Specific Comments**

C: Page 6. Triggering mechanisms. Many alternative mechanisms (other than pore pressure increase) are presented for seismicity triggers but the paper then only goes on to explore a few of these explicitly. For example, heterogeneity and geochemical effects are not discussed further. Thermal effects are considered but no detailed assessment of rock properties and contrast of layers. No further discussion of stress redistribution or aseismic slip. Thus I am left feeling the conclusion that seismicity can be predicted, monitored and managed is undermined by not tackling these in detail.

Reply: The objective of providing a detailed list of triggering mechanisms other than pore pressure build-up was to clearly show that the widespread idea that induced seismicity is exclusively caused by pore pressure increase is not accurate and that other processes should be considered. As for the completeness of non-isothermal effects, we agree that the effect of the contrast of rock properties between layers is relevant. For example, if the rock layers have different thermal expansion coefficient, the shear stress increases in the contact between the

two layers, which may result in damage to the lower portion of the caprock around injection wells.

Authors' changes: We have added a detailed explanation of the non-isothermal effect of having stiffness contrast between the storage formation and the caprock (see page 14, lines 11-22, in the revised manuscript). Additionally, in order to tackle all the triggering mechanisms, we have added two new Sections entitled "Shear slip stress transfer" and "Geochemical effects on geomechanical properties" (placed after the Section on "Non-isothermal effects"). In this way, we give details and discuss all the triggering mechanisms mentioned in Section 2.

C: Page 8. Stress state It is a very large assumption to say sedimentary rocks are not critically stressed. There are clearly many exampled (even cited in this paper, e.g. Blackpool) where sedimentary rocks are critically stressed. The last sentence of this section admits this but it does not appear valid to me to makes this strong assertion/ assumption, particularly as displayed in Figure 3.

Reply: The Section on "Stress state" started by stating that sedimentary rocks are generally not critically stressed. Sedimentary rocks are more ductile or plastic (sometimes called soft rocks) as compared, for instance, to igneous and metamorphic crystalline rocks, which behave in a more fragile manner. As sedimentary rocks have lower stiffness compared to other rocks, stress state is generally more isotropic, i.e., subject to less deviatoric stresses. By this, we are not affirming that they are never critically stressed, but that, in general, this is the case, which is favourable for $CO_2$ storage because injection is performed in sedimentary basins. To support this statement, we now provide a Table with the stress state at several $CO_2$ injection sites and the corresponding mobilized friction coefficient. We agree that if one does not read the caption, Figure 3 may give the impression that all sedimentary formations are not critically stressed, so we have modified the Figure to avoid this.

Authors' changes: We have rephrased most part of the Section on "Stress state". In particular, we have moved the last paragraph, where we stated that sedimentary rocks may be critically stressed, towards the beginning of the section. Additionally, we have included a Table with the stress state at several $CO_2$ injection sites, in which it can be seen that any of them are not critically stressed. We have also modified Figure 3 to make it clear that sedimentary rocks are less likely to be critically stressed than crystalline rocks, but that they may be critically stressed in some cases.

C: Page 9 Pressure Buildup Evolution. This may be many, and even incorrect, but is the term "pressure buildup" used correctly here? This phrase, to me, implies the early stage of injection, or the build-up to max sustained pressure. Here it used to describe what i would call 'pressure evolution' over the whole project. In the discussion of Figure 4, is this new work? How was it modelled? What are the boundary conditions, scales etc etc? Labelling on the figure also needs improved.

Reply: The described pressure evolution occurs as long as the pressure perturbation does not reach the boundaries of the aquifer. Figure 4 shows the pressure evolution for the first year of injection, but even if injection is maintained for decades e.g., 30 years, the injection pressure remains practically constant at the injection well. To avoid reaching the aquifer boundary during a 30-year injection, the radius of the model needs to be of some 100 km. In reality, we rarely find aquifers with extremely large size. If a boundary is reached by the pressure perturbation front, injection pressure will increase or decrease depending on whether the outer boundary is low-permeable or high-permeable, respectively. Beyond boundary effects, pressure tends to stabilize due to brine leakage through the caprock and base rock.

Authors' changes: We have modified the term "pressure buildup" by "pressure increase" and refer to "pressure evolution" instead of "pressure buildup evolution". We also provide an explanation of the boundary effects on pressure evolution (page 11, lines 7-13; the page and line numbers correspond to the version with track changes) and provide the necessary details of the model in the caption of Figure 4.

C: Page 10. Here the authors state that pressure dissipation can be accommodated by brine leaking through a fault but not $CO_2$. They need to be explicit as to why this is the case. e.g. in the last sentence of this section this should state there is high entry pressure 'to $CO_2$' specifically and that there is (presumably) a lower entry pressure for brine.

Reply: Since the caprock and faults are fully saturated with the resident brine, brine flow is a single phase problem, and thus, there is no entry pressure for brine flow. In addition, it should be taken into account that the pressurized area will become of several thousands of square kilometres in the long-term. Thus, even for the small flux of brine that will occur across the caprock, the total volume of displaced brine will be very large. We will explain in more detail this aspect.

Authors' changes: We now specify that the entry pressure refers to $CO_2$. Additionally, we have quantified the flow rate of brine that may leak through the caprock, effectively lowering pressure increase in the storage formation (page 12, lines 12-19).

C: Page 11. Non Isothermal Effects As with the pressure modelling, is this new work here? For example, on lines 6-10 of page 11 is this new work or results from Jeanne et al? lines 11-13. Is this a general comment or for a specific model/conditions? lines 16-18. Why? Is this because there is only cooling in the reservoir not caprock? line 19. Why especially in normal faulting regimes? line 21 - end of section. Is all this discussion (and figure 6) all based on modelling? As for above, what conditions, modelling approach etc etc if it is new.

Reply: As explained in the response to the General Comments, the content of this section is a compilation of recent work (both by the authors and by other contributors in the literature). All of these results are based on numerical modelling. Lines 6-13 provide a general explanation of thermo-mechanical effects which have been observed in our simulations and also by other authors. The explanation to the statement made in lines 16-18 is explained in the next paragraph. As it can be seen in the simulation results shown in Figure 5, cooling occurs both in the reservoir and the lower portion of the caprock. Thus, thermal stresses occur in both formations within the cooled region. However, the reduction in the vertical stress within the reservoir generates an imbalance in stress equilibrium. Similarly to what occurs in tunnel excavations, there is a stress redistribution around the cooled region, which results in an increase in the horizontal stresses in the lower portion of the caprock. This increase improves caprock stability in normal faulting stress regimes, because the deviatoric stress is reduced. However, the deviatoric stress increases in the lower portion of the caprock in reverse faulting stress regimes as a result of this stress redistribution. We now provide a more detailed explanation of these processes and their implications.

Authors' changes: We now explain in the manuscript the relevant conditions to understand the modelling results. We have added Figure 5a with the model setup, including the initial and boundary conditions, and we have added Table A1 in the Appendix including the material properties of the simulation results. We have written a new paragraph (page 15, lines 1-13) explaining the stability changes in the caprock in strike slip and reverse faulting stress regimes.

C: Page 14 Fault Stability. line 6-8. Surely depends on the orientation of the strata (if in sed rocks) relative to the well, not that the well is horizontal? line 24-25. What does 'more deformable' mean? Is this a condition set in the model? page 15 line 8. Why is reservoir stiffer? Is this a condition of the model again?

Reply: What we mean by horizontal well is that it has a long open section, i.e., more than 1 km, like the wells at In Salah, Algeria. If the storage formation has a slope of some degrees, a "horizontal" well should follow that inclination. Thus, we agree that the proper term is 'parallel to the strata' rather than 'horizontal well'. By 'more deformable' we mean that the Young's modulus is lower. The modelling results presented in this Section use properties measured in the laboratory. For the reservoir, we consider the properties of Berea sandstone and for the caprock and base rock we consider the properties of Opalinus clay. This is why the reservoir is stiffer than the base rock. We now provide more details on the model.

Authors' changes: We have changed 'horizontal well' by 'parallel to the strata'. We now provide all the materials properties of the model shown in this Section in Tables A2 and A3 of the Appendix.

C: Characterization techniques pg 17 line 1. Stress orientations and magnitudes are pretty hard to measure from core. Can this be changed to 'most properties'. pg 18 line 2. Do we need to be careful here about formation/caprock damage here? How is this different/beneficial to say a XLOT in the caprock? pg 19 line 1. Heterogeneity is the crucial bit here. I'm not sure you can confidently infer the next section (and figure 10) when heterogeneity could easily give the same results.

Reply: In the lab measurements from cores, we were referring to the hydraulic, thermal and geomechanical properties of the rocks. Since the sentence in p. 17 line 1 may lead to confusion, we have rephrased it. As for the potential damage to the caprock, if microseismicity is induced in the caprock, shear slip of fractures may enhance permeability (by one to two orders of magnitude according to lab rock experiments), but most importantly, may reduce $CO_2$ entry pressure. Thus, it is preferable to limit microseismicity in the caprock. Nevertheless, the amount of assumable damage could vary site specifically. For example, the caprock thickness at In Salah, which was of several hundreds of metres, may allow to accept some damage to the lower portion (some meters) of the caprock because the overall caprock integrity will not be compromised. XLOT should be done in the caprock to estimate the stress state, but the maximum sustainable injection pressure will be always lower than the fracturing pressure.

Authors' changes: We have rephrased the sentence regarding characterization from cores to "Hydraulic, thermal and geomechanical properties of rock can be measured in the laboratory from core samples or in the field.".

C: Minimising Risk. pg 21. line 16 onwards. This section/bulletpoint seems a little out of place here. Sure, co-injection etc. could be used but there are other ways to manage pressure too (from straight water production and disposal to changing injection rate, WAG or not etc etc) and for a section entitled other storage concepts there are lots of other methods (basalt storage e.g.). The link to geothermal energy seems out of place/unnecessary.

Reply: With this bullet point we wanted to highlight that fluids, either brine or $CO_2$, can be produced to lower pressure build-up. The intention was not to be an exhaustive review of all proposed methods. And we mention these two alternatives as examples.

C: Figures 4-8 in particular need more scale bars, description of colours used etc. Fig 5 in particular needs better labelling to show which Mohr diagram is for which layer.

Reply: Figures 5-8 already include the scale bar and colour description. The location of the Mohr circles shown in Figure 6 is indicated in the insets of both Figures 6a and 6b, but more details on the exact location of the points will be stated in the caption.

Authors' changes: We have added the scale bar to Figure 4. We have also adapted the colour scale in Figure 6a. We now provide the exact location of the Mohr circles in the caption of Figure 6.

**REFEREE #2**

**General Comments**

C: The authors attempt to mitigate undesirable induced seismicity by investigating different mechanisms leading to fracture/fault instability and performing numerical simulations. The authors mention that the main factors causing stress changes in the reservoir are injection-related pressure buildup, in-situ stress state, injected fluid's temperature gradient. The outline of the paper is communicated at the end of Section 1 in page 4. However, there is no clear section on what unique contributions this study is making to improve the state-of-the-art. A general theme of the manuscript is that too many generic, qualitative comments are made without new data or analysis to support those comments. There is an unreasonably large emphasis on citing and reviewing existing papers instead of showing new results. When the simulation results are shown, there are no clear quantitative details of the simulation model: model dimensions, meshing, initial and boundary conditions, well conditions, and hydraulic/mechanical properties. This suggests that the manuscript should be submitted as a review article, not Research Article.

Reply: As we explained in the response to the general comment of referee #1, this is a review article, because as awardee of the Outstanding Early Career Scientists Award for the Division on Energy, Resources and the Environment (ERE) of the EGU, I was invited to publish a paper in one of the EGU journal based on my lecture. Since I presented in my lecture the work that I have done in the last years and that contributed to receive the award, the article type should be changed from research article to review article. We apologize for this mistake when we submitted the manuscript.

**Specific Comments**

C: Figure 1,2,3: They are extremely generic, redundant and partially inaccurate. For example, Figure 2 shows that the effect of temperature change is to only shift the Mohr Circle to left, which is highly imprecise and can be inaccurate depending on the rock type, injection layer geometry (total stress can change), and the magnitude and direction of temperature change. Figure 3 lumps all sedimentary rocks in the world as critically unstressed and assumes that they all fail under linear Mohr Coulomb condition. This is almost unscientific and completely unnecessary.

Reply: These three Figures are schematic to explain general aspects of induced seismicity. Regarding the shift of the Mohr circle due to temperature change, it is shifted to the left because cooling is expected to occur around $CO_2$ injection wells, and thus, a total stress reduction will occur. We have added a minus in front of the delta T to indicate that cooling takes place. Additionally, the size of the Mohr circle changes because the changes in the total stresses may

be different in the vertical and horizontal directions. Nonetheless, it may be difficult to observe that the two circles (the red and the blue ones) have different sizes in the original Figure, so we have modified it to exaggerate this effect. As for Figure 3, we agree with the referee that not all sedimentary rocks are not critically stressed, as we already state in the figure caption and main text. We also agree with the referee that the failure envelope is not linear for rock. Indeed, we usually use non-linear shear strength in some of our studies. Since the Figure was schematic, we were just representing a linear failure surface, but we have modified it to show the non-linearity of shear strength. Additionally, we now indicate in this Figure that crystalline rock is more likely to be critically stressed than sedimentary rocks because of their higher stiffness, which makes them accumulate more stress. Additionally, to support this statement, we have added Table 1 showing the stress state at several $CO_2$ storage sites together with the mobilized friction coefficient. The mobilized friction coefficient ranges from 0.35 to 0.54, so in all cases is lower than 0.6, meaning that favourably oriented faults to undergo shear slip are not critically stressed. Of course, knowing the stress state at each site is crucial because the maximum sustainable injection pressure to avoid reactivating faults depends on the initial stress of state. Thus, the pressure increase at the site with a mobilized friction coefficient of 0.54 has to be lower than that at the site with a mobilized friction coefficient of 0.35.

Authors' changes: We have modified Figures 1-3 to clarify the points raised by the referee.

C: Figure 4: This shows results for a problem that is not even defined. What is the physical model setup, what are the initial and boundary conditions of the coupled flow-mechanics problem, what is the well rate and injection duration? Why do we accept this result as correct?

Reply: This Figure describes the pressure evolution in a 100-m thick reservoir in which 1 Mt of $CO_2$/yr are injected in an aquifer with permeability of 1e-13 m2 and radius of 100 km. Since the pressure front does not reach the outer boundary during the injection period shown in the figure, the nature of the boundary does not have any effect on the pressure evolution. The aquifer, which is placed at 1.5 km depth, initially presents hydrostatic pressure. Nevertheless, since we show the pressure changes, the absolute initial pressure is not relevant. Regardless of the particularities of this model, the intention is to describe in a general way $CO_2$ injection pressure evolution, which is significantly different from that of water injection. As explained in the text, the characteristics of this pressure evolution, i.e., the initial sharp increase in $CO_2$ pressure followed by a relatively constant injection pressure, have been observed in the field, in analytical and numerical solutions. Based on this evidence, it can be accepted as correct.

Authors' changes: We now explain the model details in the caption of Figure 4.

C: Figure 5: Same as before. Why is this an accepted solution? What is the problem setup?

Reply: The results shown in this figure are from a fully coupled numerical code that solves non-isothermal two-phase flow in deformable porous media (CODE_BRIGHT), which has been extensively benchmarked and is well accepted within the scientific community. Nevertheless, the Figure was intended to support the explanations of the processes that occur during cold $CO_2$ injection, without focusing on a specific case.

Authors' changes: We have included the model setup in Figure 5a and the material properties in Table A1 in the Appendix.

C: Page 9: "progressively increasing the flow rate at the beginning of injection may avoid the initial peak in pressure buildup" This statement needs to be quantified: how much increase to

avoid how much pressure buildup. Otherwise, the idea of "progressively increasing the rate" is a conjecture.

Reply and Authors' changes: We have deleted this sentence.

C: Page 1-15: There is too much literature review. Almost 90.

Reply: We deem this amount of references appropriate for a review article.

C: Abstract: "We aim at understanding … and to develop methodologies … through dimensional and numerical analysis." There is now dimensional analysis. In fact, the word "dimensional" appears only once in the abstract. Please remove it from the abstract.

Reply and Authors' changes: We have removed the word dimensional in the abstract.

C: Page 14-15: This combines citations with discussion of authors' results. This is very confusing. It is better to move authors' own work into a separate section and not mix with background literature survey.

Reply: In this section, we are providing explanations of the relevant aspects that control fault stability. Even though we have studied this problem extensively, other authors have made relevant contributions to the topic and we believe that it is important to include their contributions in this section as well.

C: Page 15 line 5: "As a result, the induced horizontal stresses in the in-plane direction are high where the storage formation is present on both sides of the fault, but it is low where the base rock is on the other side of the fault." This is not a result in this manuscript. Either remove it or support it with actual simulation results.

Reply: This statement results from the observation of the changes in the horizontal stress in the in-plane direction shown in Figure 7.

Authors' changes: We have added a reference to Figure 7 at the end of the sentence.

C: Figure 7 and 8: Data used for the simulation must be provided otherwise it is not clear what to expect in the result. What is the contrast in elastic stiffness and hydraulic properties between the damage zone vs. reservoir vs. caprock. All modeling assumptions used during the simulation must be listed.

Reply and Authors' changes: We now provide more details on the model (page 21, lines 16-19) that complement the information already provided (page 21, lines 8-16) and include the material properties in Tables A2 and A3 in the Appendix.

C: Page 17-18: This proposes a field test to macroscopically characterize hydraulic, thermal and geomechanical properties without mentioning any challenges related to applicability and operation. Otherwise such a field test will get classified as unrealistic and not useful for $CO_2$ injection.

Reply: We thank the referee for raising this point, which is certainly of interest and deserves discussion. There are a number of challenges related to this characterization test. To begin with, the drilling of a network of monitoring wells is not yet common practice. Monitoring techniques

also present challenges. Pressure is usually measured at the well-head, but calculating the bottom-hole pressure from the well-head pressure is not straightforward given the non-linearities of the injected fluid, especially for $CO_2$ injection. Unfortunately, pressure measurements in wells different from the injection well are almost non-existent. Temperature measurements receive even less attention. As for deformation measurements, ground surface deformations can be measured with InSAR data, but for characterization tests that last a few days, the deformation of the ground may not be detectable given the great depths of storage formations. Thus, deformation should be measured at depth within the boreholes. These measurements pose the question of whether the measured deformation refers to that of the rock or to that of the well. Since the casing of wells is stiffer than rock, the rock may deform more than the well and sliding could even occur between the rock and the cement surrounding the well casing. Fiber optic may solve part of these monitoring challenges, but the way how this monitoring should be performed is still not crystal clear for the moment. As far as microseismicity monitoring is concerned, arrays of geophones are certainly needed to be placed at depth. Otherwise, the signal-to-noise ratio is too high, which complicates detecting microseismic events. Additionally, multi-sensor arrays with a wide aperture coverage are necessary to accurately locate the events.

Authors' changes: We now include in the manuscript this discussion on the challenges of performing such characterization test (page 26, lines 12-25 and page 27, lines 1-7).

C: Page 21: "predictive models of induced seismicity that consider coupled THMS processes should be applied" This is much easier said than done. What are these models? The results in this manuscript do not show any coupling to seismicity, which requires solution of the elastodynamic problem in a n-dimensional domain with a (n-1) dimensional fault surface, not a n-dimensional fault zone. This manuscript presents neither an approach nor results from coupling of the four processes T, H, M, S.

Reply: This is a recommendation we made for future practices based on our previous experience. Given that we do not go into the details of the seismic part, we will replace THMS by THM, which is discussed in the manuscript.

Authors' changes: We have removed the seismic part.

C: Page 21: "The continuous characterization will permit updating the fault stability analysis by incorporating newly detected faults." How will the new faults be detected? This is not trivial and not answered in this manuscript. So, please remove this.

Reply: The continuous characterization refers to the methodology explained in Figure 10. Thus, by applying this methodology, it is possible to detect previously unidentified low-permeable faults and incorporate them in the model of the injection site.

Authors' changes: We now mention Figure 10 at the end of this sentence to clarify how new faults can be detected.

C: Figure 6: Color scale can be improved. For example, it is different for the upper and lower figures, yet the maximum value is not visible in the upper figure.

Reply and Authors' changes: We have modified the colour scale of Figure 6a so that the maximum and minimum values are visible.

[revised manuscript text omitted]
 20 ºC through a vertical well. While (bc) and (ed) are plotted at the same scale, (ab) is plotted at a smaller scale.

[Figure]

Figure 6: Total stresses in the (a) vertical and (b) horizontal direction after 2 years of injecting 0.2 Mt/yr of $CO_2$ at 20 ºC through a vertical well, indicating the sign of the induced stresses. Thermal stresses, $\Delta\sigma_T$, are proportional to the bulk modulus, $K$, the thermal expansion coefficient, $\alpha_T$, and the temperature difference, $\Delta T$. The changes in the Mohr circles at a point placed 25 m away from the injection well in (c) the reservoir (2 m below the reservoir-caprock interface) and (d) the caprock (2 m above the reservoir-caprock interface) are  also represented.

[Figure]

Figure 7: (a) Geological setting in a normal faulting stress regime (plane strain model), including a low permeability fault that leads to (b) reservoir pressurization, $\Delta P$, and (c) horizontal total stress changes in the in-plane direction, $\Delta\sigma_x$, when $CO_2$ is injected in the hanging wall at a rate of $2\cdot10^{-3}$ kg/s/m for 1 year.

[Figure]

Figure 8: Distribution of stability changes induced by the pressure and stress changes shown in Figure 7, measured in terms of the mobilized friction angle changes, $\Delta\phi_{mob}$. The inset shows the Mohr circles before and after reservoir pressurization.

[Figure]

Figure 9: Hydro-mechanical characterization test proposed by Vilarrasa et al. (2013c) to quantify the rock properties at the field scale and obtain an initial estimate of the maximum sustainable injection pressure. *P* refers to pressure, *T* to temperature and $u_z$ to vertical displacement.

[Figure]

Figure 10: (a) Concept of the continuous characterization technique proposed by Vilarrasa et al. (2017c) to detect and locate low permeability faults using diagnostic plots; (b) asymmetric $CO_2$ plume as a result of the additional pressurization caused by a low-permeability fault, which displaces $CO_2$ towards the opposite direction of the fault; (c) detection of multiple faults by updating the conceptual model of the site and comparing field measurements with predictive simulations; and (d) estimation of the

fault location from the measured divergence time in the derivative of the pressure evolution using type curves.